# The development of active binocular vision under normal and alternate rearing conditions

Lukas Klimmasch[1]*, Johann Schneider[1], Alexander Lelais[1], Maria Fronius[2], Bertram Emil Shi[3], Jochen Triesch[1]*

[1]Frankfurt Institute for Advanced Studies (FIAS), Frankfurt am Main, Germany; [2]Department of Ophthalmology, Child Vision Research Unit, Goethe University, Frankfurt am Main, Germany; [3]Department of Electronic and Computer Engineering, Hong Kong University of Science and Technology, Hong Kong, China

**Abstract** The development of binocular vision is an active learning process comprising the development of disparity tuned neurons in visual cortex and the establishment of precise vergence control of the eyes. We present a computational model for the learning and self-calibration of active binocular vision based on the Active Efficient Coding framework, an extension of classic efficient coding ideas to active perception. Under normal rearing conditions with naturalistic input, the model develops disparity tuned neurons and precise vergence control, allowing it to correctly interpret random dot stereograms. Under altered rearing conditions modeled after neurophysiological experiments, the model qualitatively reproduces key experimental findings on changes in binocularity and disparity tuning. Furthermore, the model makes testable predictions regarding how altered rearing conditions impede the learning of precise vergence control. Finally, the model predicts a surprising new effect that impaired vergence control affects the statistics of orientation tuning in visual cortical neurons.

*For correspondence:
klimmasch@fias.uni-frankfurt.de
(LK);
triesch@fias.uni-frankfurt.de (JT)

**Competing interests:** The authors declare that no competing interests exist.

## Introduction

Humans and other species learn to perceive the world largely autonomously. This is in sharp contrast to today's machine learning approaches (*Kotsiantis et al., 2007*; *Jordan and Mitchell, 2015*), which typically use millions of carefully labeled training images in order to learn to, say, recognize an object or perceive its three-dimensional structure. How can biological vision systems learn so much more autonomously? The development of binocular vision presents a paradigmatic case for studying this question. This development is an active process that includes the learning of appropriate sensory representations and the learning of precise motor behavior. Species with two forward facing eyes learn to register small differences between the images projected onto the left and right retinas. These differences are called binocular disparities and are detected by populations of neurons in visual cortex (*Kandel et al., 2000*; *Blake and Wilson, 2011*) that have receptive subfields in both eyes. Frequently, they are modeled using separate Gabor-shaped filters for each eye, where the disparity is encoded by a shift in the centers of the filters, a difference between their phases, or by a combination of both (*Fleet et al., 1996*; *Chen and Qian, 2004*). The responses of such disparity tuned neurons can be used to infer the three-dimensional structure of the world. At the same time, such species also learn to align their eyes such that the optical axes of the two eyes converge on the same point of interest. These so-called vergence eye movements are also learned and fine-tuned during development (*Held et al., 1980*; *Fox et al., 1980*; *Stidwill and Fletcher, 2017*). Again, this learning does not require any supervision from outside, but must rely on some form of self-calibration.

Various computational models have been employed to explain the development of binocular disparity tuning in the context of efficient coding ideas (*Li and Atick, 1994*; *Hunt et al., 2013*), independent component analysis (ICA) (*Hoyer and Hyvärinen, 2000*), Bayesian inference (*Burge and Geisler, 2014*), or nonlinear Hebbian learning (*Chauhan et al., 2018*) (see *Chauhan et al., 2020* for a review). A critical limitation of these studies is that they ignore the importance of behavior in shaping the statistics of the sensory input and in particular the role of vergence eye movements in determining the statistics of disparities. Indeed, while it has long been argued that the development of disparity tuning and vergence eye movements are interdependent (*Hubel and Wiesel, 1965*; *Candy, 2019*), it has been only recently that computational models have tried to explain how the learning of disparity tuning and vergence eye movements are coupled and allow the visual system to self-calibrate (*Zhao et al., 2012*; *Klimmasch et al., 2017*; *Eckmann et al., 2019*). These models have been developed in the framework of Active Efficient Coding (AEC), which is an extension of Barlow's classic efficient coding hypothesis to active perception (*Barlow, 1961*). In a nutshell, classic efficient coding argues that sensory systems should use representations that remove redundancies from sensory signals to encode them more efficiently. Therefore, sensory representations should be adapted to the statistics of sensory signals. Based on this idea, a wide range of data on tuning properties of sensory neurons in different modalities have been explained from a unified theoretical framework (*Dan et al., 1996*; *Vinje and Gallant, 2000*; *Simoncelli, 2003*; *Smith and Lewicki, 2006*; *Doi et al., 2012*). AEC goes beyond classic efficient coding by acknowledging that developing sensory systems shape the statistics of sensory signals through their own behavior. This gives them a second route for optimizing the encoding of sensory signals by adapting their behavior. In the case of binocular vision, for example, the control of vergence eye movements is shaping the statistics of binocular disparities. By simultaneously optimizing neural tuning properties and behavior, AEC models have provided the first comprehensive account of how humans and other binocular species may self-calibrate their binocular vision through the simultaneous learning of disparity tuning and vergence control.

A generic AEC model has two components. The first component is an efficient coding model that learns to encode sensory signals by adapting the tuning properties of a population of simulated sensory neurons (*Olshausen and Field, 1996*; *Olshausen and Field, 1997*). In the case of binocular vision, this is a population of visual cortical neurons receiving input from the two eyes that learns to encode the visual signals via an efficient code. The second component is a reinforcement learning (RL) model that learns to control the behavior. In the case of binocular vision, this component will learn to control eye vergence. For this, it receives as input the population activity of the visual neurons and learns to map it onto vergence commands. This learning is guided by an internally generated reward signal, which reinforces movements that lead to a more efficient encoding of the current visual scene. For example, when the eyes are aligned on the same point, the left and right images become largely redundant. The efficient coding model can exploit this redundant structure in both eyes, by developing neurons tuned to small or zero disparities. Conversely, such binocular neurons tuned to small disparities will represent any remaining misalignments of the eyes, providing informative input for vergence control. In this way, learning of vergence control supports the development of neurons tuned to small disparities and this developing population of neurons in turn facilitates the learning of fine vergence control (*Zhao et al., 2012*).

Importantly, however, this normal development of binocular vision is impaired in a range of alternate rearing conditions. In fact, already since the days of Hubel and Wiesel, alternate rearing conditions have been used to improve our understanding of visual cortex plasticity and function. Manipulating the input to the visual system during development and observing how the system reacts to such manipulations has shaped our understanding of visual development until today. For example, artificially inducing a strabismus (squint) leads to drastic changes in the tuning properties of neurons in visual cortex (*Hubel and Wiesel, 1965*). A comprehensive theoretical account of the development of binocular vision must therefore also be able to explain the experimentally observed differences in alternate rearing conditions. Our recent work (*Eckmann et al., 2019*) took a step in this direction by modeling the development of amblyopia in response to anisometropic rearing (introducing differences between the refractive power of the two eyes). In the present study, in contrast, we aim to demonstrate the generality of the AEC approach by reproducing and explaining a large range of neurophysiological findings from different alternate rearing conditions: changing the orientation distribution in the visual input (horizontal, vertical, or orthogonal rearing), monocular

rearing, strabismic rearing, and aniseikonia. We also utilize a sophisticated biomechanical model of the oculomotor system, opening the door to simulating the effects of both optical and motor aberrations on visual development.

Our results show that the model qualitatively captures findings on how different alternate rearing conditions alter the statistics of disparity tuning and binocularity. Furthermore, the model makes specific novel and testable predictions about differences in vergence behavior under the different rearing conditions. Surprisingly, it also predicts systematic differences in the statistics of orientation tuning of visual cortical neurons depending on the fidelity of vergence eye movements. Overall, our results support AEC as a parsimonious account of the emergence of binocular vision, highlighting the active nature of this development.

## Results

### A model for the development of active binocular vision

The model comprises a virtual agent situated in a simulated environment. The agent looks at a textured plane that is positioned in front of it at variable distances between 0.5 m and 6 m (*Figure 1A*). We use planar images instead of a full 3D environment to (i) allow us to uniquely define the correct vergence angle for the current visual scene and (ii) make sure that the visual input follows natural image statistics. Note, that previous AEC models have already demonstrated the approach in full 3-D environments (*Zhu et al., 2017a*; *Zhu et al., 2017b*; *Lelais et al., 2019*).

An image is rendered for the left eye and a second image is rendered for the right eye. Binocular patches are extracted from these images and encoded by a sparse coding algorithm. The activation levels of the learned binocular basis functions (BFs) can be thought of as firing rates of binocular simple cells in primary visual cortex. The basis functions themselves roughly describe their receptive fields (RFs) and are adapted through learning (*Olshausen and Field, 1997*). These activations are then squared and pooled across the image to obtain a more position-invariant representation mimicking the behavior of complex cells. From this state representation a reinforcement learner generates vergence commands that symmetrically rotate the eyeballs inwards or outwards. This results in two new images being rendered and a new simulation iteration starts. The complete process is depicted in *Figure 1B* (see Materials and methods for details).

In the human retina, the RF size of ganglion cells increases towards the periphery (*Curcio et al., 1990*). We incorporate this idea by extracting patches from an input image at two different spatial scales: A high-resolution fine scale is extracted from the central part and a low-resolution coarse scale is extracted from a larger area (orange and turquoise boxes in *Figure 1* and *Figure 2*). The

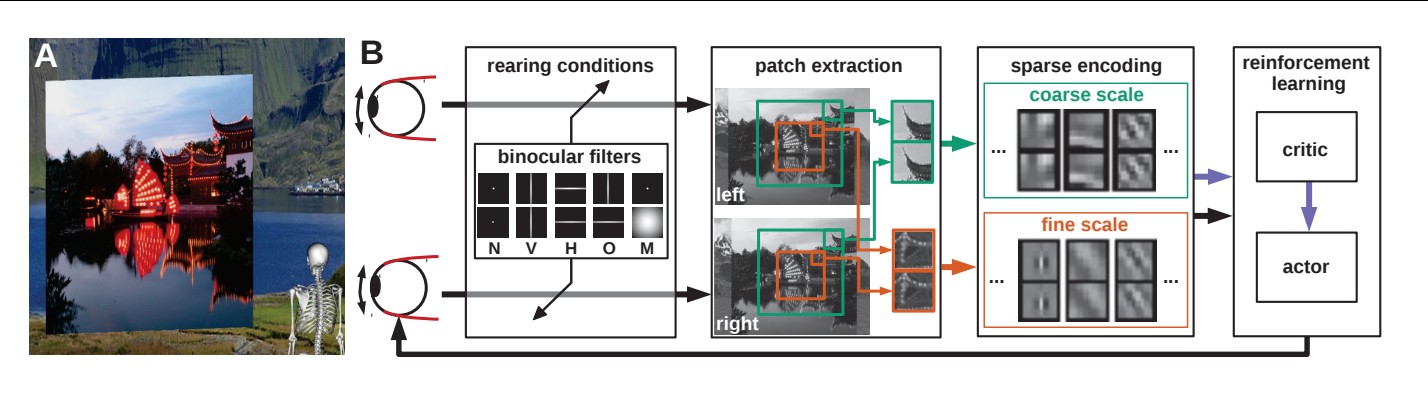

**Figure 1.** Model overview. (A) The agent looks at the object plane in the simulated environment. (B) Architecture of the model. Input images are filtered to simulate alternate rearing conditions (N: normal, V: vertical, H: horizontal, O: orthogonal, M: monocular). Binocular patches are extracted at a coarse and a fine scale (turquoise and orange boxes) with different resolutions. These patches are encoded via sparse coding and combined with the muscle activations to form a state vector for actor critic reinforcement learning. The reconstruction error of sparse coding indicates coding efficiency and serves as a reward signal (purple arrow) to train the critic. The actor generates changes in muscle activations, which result in differential rotations of the eyeballs and a new iteration of the perception-action cycle.

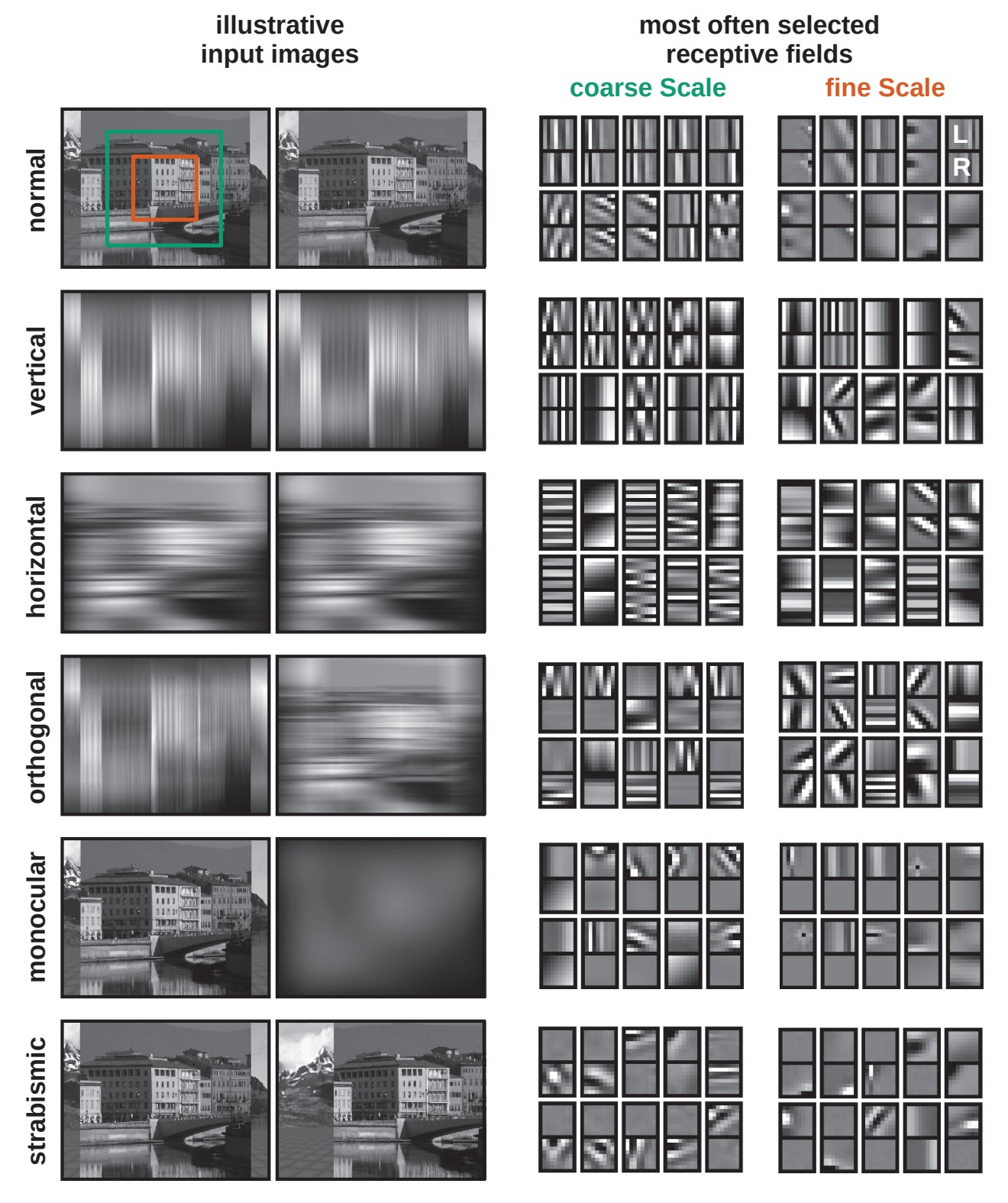

**Figure 2.** Visual input and learned receptive fields under different rearing conditions. Left: Illustration of visual inputs under the different rearing conditions. Except for the *normal* condition, the images are convolved with different Gaussian filters to blur out certain orientations or simulate monocular deprivation. To simulate strabismus the right eye is rotated inward by 10°, so that neurons receive non-corresponding inputs to their left and right eye receptive fields. The structures behind the object plane depict a background image in the simulator. Right: Examples of binocular RFs for the

*Figure 2 continued on next page*

*Figure 2 continued*

fine and coarse scale learned under the different rearing conditions after 0.5 million iterations. For each RF, the left eye and right eye patches are aligned vertically. In each case, the 10 RFs most frequently selected by the sparse coding algorithm are shown.

The online version of this article includes the following source data and figure supplement(s) for figure 2:

**Source data 1.** Coarse and fine scale RFs in vectorized form for all rearing conditions.

**Figure supplement 1.** All learned RFs for the six rearing conditions.

overlap between the coarse and fine scale does not depict the biological reality, but simplifies the implementation and analysis of the model. Covering a visual angle of 8.3° in total, the fine scale corresponds to the central/para-central region (including the fovea) and the coarse scale to the near-peripheral region with a diameter of 26.6°. On the one hand, this two-scale architecture is more biologically plausible than using just a single scale, on the other hand it also increases the resulting verging performance (*Lonini et al., 2013*). One input patch (or subfield) in the coarse scale can detect a disparity of up to 8.8° while one patch in the fine scale covers 1.6°. The coarse scale can therefore be used to detect large disparities, while the fine scale detects small disparities.

We simulate altered rearing conditions by convolving the input images for the two eyes with two-dimensional Gaussian kernels to blur certain oriented edges, or to simulate monocular deprivation. To mimic strabismus, the right eyeball is rotated inwards while the left eye remains unchanged to enforce non-overlapping input to corresponding positions of the left and right retina (see Materials and methods for details).

The adaptation of the neural representation and the learning of appropriate motor commands occur simultaneously: While the sparse coder updates the initially random RFs to minimize the reconstruction error, the RL agent generates vergence eye movements to minimize the reconstruction error of the sparse coder. Since the sparse coder has a fixed capacity, minimizing its reconstruction error is equivalent to maximizing its coding efficiency. Thus, both the sparse coder and the reinforcement learner aim to maximize the overall coding efficiency of the model. The learning of the two components (sparse coder and RL agent) happens roughly at the same timescale. Our model is robust to variations in the learning rates, as long as the reinforcement learner's critic converges faster than the actor (*Van Hasselt and Wiering, 2007*).

## Normal rearing conditions lead to the autonomous learning of accurate vergence control for natural input and random dot stereograms

A first critical test of a model of the development of binocular vision is whether the model produces plausible behavior. Indeed, under normal rearing conditions the joint learning of the neural representation and motor behavior results in an agent that accurately verges the eyes on the plane in front of it (*Klimmasch et al., 2017*). *Video 1* illustrates the learned behavior.

To quantify vergence behavior in the model, we define the absolute vergence error. It measures by how much the vergence angle between the eyes deviates from the ideal position at the end of a fixation (see Materials and methods for details). The model obtains an accuracy of vergence eye movements of 0.12 ± 0.17° or 455.40 ± 613.75 arc sec. Note, however, that the model as described above has a much lower visual resolution compared to human foveal vision. One pixel in the model corresponds to 802 arc sec, while the spacing between photoreceptors in the fovea corresponds to 28 arc sec. When we correct for the model's lower visual resolution (see Materials and methods), the corrected vergence

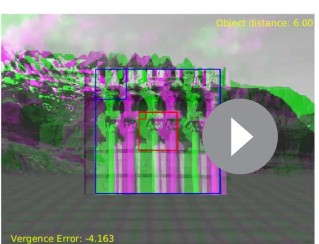
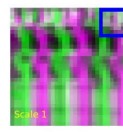
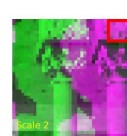

**Video 1.** Vergence performance for normal visual input. The sizes of the scales and the according patch sizes are indicated in blue for the coarse scale and red for the fine scale.

https://elifesciences.org/articles/56212#video1

accuracy is 15.9 ± 21.44 arc sec. This falls within the range of human performance under natural viewing conditions, which is typically better than 60 arc sec (1 arc min) for stimuli not closer than 0.5 m (*Jaschinski, 1997*; *Jaschinski, 2001*).

A second critical test of a model of the development of stereoscopic vision is whether it can handle *random-dot stereograms* (RDSs), which represent the most challenging stimuli for stereopsis (*Lee and Olshausen, 1996*; *Chen and Qian, 2004*; *Chauhan et al., 2018*). Since their introduction by *Julesz, 1971* RDSs have been used extensively to investigate the human ability for stereoscopic vision. Nowadays, they are used in opthalmological examinations to asses stereo acuity as well as to detect malfunctions in the visual system, such as strabismus or amblyopia (*Walraven, 1975*; *Okuda et al., 1977*; *Ruttum, 1988*). In these experiments, participants view a grid of random dots through a stereoscope or another form of dichoptic presentation. Typically, the central part is shifted in one of the two images which results in the perception of stereoscopic depth in healthy subjects. The advantage of this form of examination is that there are no monocular depth cues (such as occlusion, relative size, or perspective). The impression of depths arises solely because of the brain's ability to integrate information coming from the two eyes.

To show that our model is able to perceive depth in RDS, although not having been trained on them, we generate various RDS and render the shifted images for the left and right eye separately. We expose the model that was trained on natural input stimuli to a range of RDS with different spatial frequencies, window sizes, disparities, and object distances. The model is able to exploit the differences in the images and align the eyes on the virtual plane that will appear in front or behind the actual object plane in the RDS. Averaged over all trials, the model achieves an absolute vergence error of 0.21 ± 0.22° at the end of a fixation. This corresponds to a corrected vergence accuracy of 26.8 ± 28.8 arc sec. This is only slightly worse than the model's performance on natural images (see Figure 6) and demonstrates that the model generalizes to artificial images it has never seen before. A video of the performance can be found in *Video 2*.

## Altered rearing conditions cause qualitative changes in neural representations

A third critical test of any model of the development of binocular vision is whether it can account for the effects of alternate rearing conditions observed in biological experiments. We simulate such alternate rearing conditions by filtering the input images for the left and right eyes with Gaussian filters. The amount of blur was chosen to simulate experiments where animals where exposed to just one single orientation during development (*Stryker et al., 1978*; *Tanaka et al., 2006*). *Figure 2* shows illustrative examples of the filtered images that were used to train our model and the respective learned RFs. Here, we only depict the 10 RFs that have been selected most often during training for each scale. The full set of all RFs can be found in *Figure 2—figure supplement 1*.

When the model is trained with unaltered natural visual input, the resulting RFs resemble Gabor wavelets (*Daugman, 1985*), as shown in the first row in *Figure 2*. The changes that are applied to the input images in the alternate rearing conditions are reflected in the RFs that are learned: Among the 10 most often selected RFs there are no vertically (horizontally) oriented RFs, when the model is trained on images that are deprived of vertical (horizontal) edges. Orthogonal RFs emerge as a result of training on orthogonal input. When one eye is deprived of input, the RFs will become *monocular* and encode information coming from the 'healthy' eye only. Strabismic rearing results in the development of monocular RFs without a preference for one or the other eye (*Hunt et al., 2013*). In the following sections, we will quantify neurons' tuning properties for different rearing conditions and compare them to neurophysiological findings.

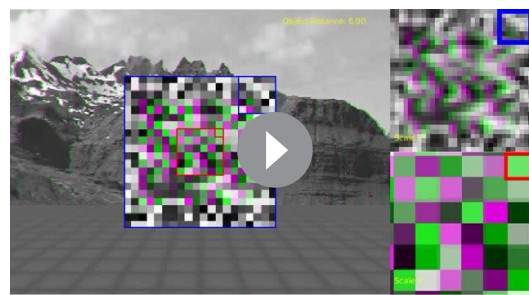

**Video 2.** Vergence performance on a randomly generated set of RDS.
https://elifesciences.org/articles/56212#video2

## Neurons' orientation tuning reflects input statistics

To analyze the statistics of the developing RFs in greater detail, we fit oriented two-dimensional Gabor wavelets to each RF (see Materials and methods for details). For this part of the analysis, the left and right parts of the binocular RFs are studied separately, that is, we consider the *monocular* RF fits only. We combine the results from coarse and fine scale, since a two-sample Kolmogorov-Smirnov test (*Young, 1977*) did not reveal a statistically significant difference between the distributions of orientation preferences (p-values > 0.18 for all rearing conditions). Only those RFs which met a criterion for a sufficiently good Gabor fit are considered for further analysis (98% of all bases, see Materials and methods for details).

*Figure 3* shows how the altered input changes the distribution of preferred orientations of the RFs. The *normal* case exhibits a clear over-representation of vertically (0°) and horizontally (90°) tuned RFs. This over-representation has been observed in different animals (*Appelle, 1972*; *Li et al., 2003*) and humans (*Furmanski and Engel, 2000*) and is considered a neural correlate of the *oblique effect*. This phenomenon describes the relative deficiency in participants' performance in perceptual tasks on oblique contours as compared to the performance on horizontal or vertical contours (*Appelle, 1972*). It has been argued that it stems from the over-representation of vertical and horizontal edges in natural images (*Coppola et al., 1998*) and reflects the internal model of orientation distribution in the world (*Girshick et al., 2011*; *Burge and Geisler, 2014*). Furthermore, it may reflect aspects of the imaging geometry (*Rothkopf et al., 2009*; *Straub and Rothkopf, 2021*). Additionally, we cannot exclude the possibility that it is related to the rectangular pixel grid representing the input to our model.

While the distribution of orientations does not change much in the *monocular* and *strabismic* rearing case, we observe a marked difference to the normal case when certain orientations are attenuated in the input. The models trained on *vertical* input are missing the peak at horizontal orientations and vice versa for the *horizontal* case. Additionally, we find an increased number of neurons tuned to the dominant orientation in the input. These observations are consistent with animal studies (*Stryker et al., 1978*; *Tanaka et al., 2006*).

The separate analysis of the RFs in the left and right eye for the models that were trained on *orthogonal* input reveals the adaptation of each eye to its input statistics. Furthermore, we find that orthogonal RFs developed (also see fourth row in *Figure 2*) that have been observed in an orthogonal rearing study in cats (*Leventhal and Hirsch, 1975*).

## The development of binocular receptive fields requires congruent binocular input

Another interesting feature of the neural representation that has been studied extensively in the context of alternate rearing is the *binocularity*. The binocularity index (BI) is used to assess how responsive a neuron is to inputs from the two eyes. A *binocular* neuron requires input from both eyes to respond maximally, while a *monocular* neuron is mostly driven by just one eye. To determine the binocularity indices for the neurons in our model, we use an adaptation of the original method from *Hubel and Wiesel, 1962* (see Materials and methods for details). The binocularity index can vary from −1 (monocular left) over 0 (binocular) to +1 (monocular right).

*Figure 4* depicts the binocularity distributions for the coarse and the fine scale for all rearing conditions. The models that were trained on input that is coherent between the left and right eye (top row) exhibit the majority of neurons falling in the bin with binocularity index 0. Neurons in this category receive about the same drive from the left and the right eye. In the *normal* case, more neurons fall into that bin than in the *vertical* and *horizontal* case. This is due to the ability of the model to perform precise vergence control: Since left and right image are almost identical most of the time, the great majority of basis functions will develop to encode the exact same input from both eyes. This, in turn, will result in the cells being completely binocular with a binocularity index of 0. This effect is even more pronounced at the coarse scale, where small residual disparities can no longer be resolved. In the vertical and horizontal rearing case, we observe a reduction in the number of cells that have a binocularity index around 0. We attribute this to the limited vergence performance in these cases, that we will analyse in the next sections.

If, however, the input differs qualitatively for the two eyes (*Figure 4*, bottom row) the receptive fields will also differ in their monocular sub-parts. This can also be observed in *Figure 2* for the

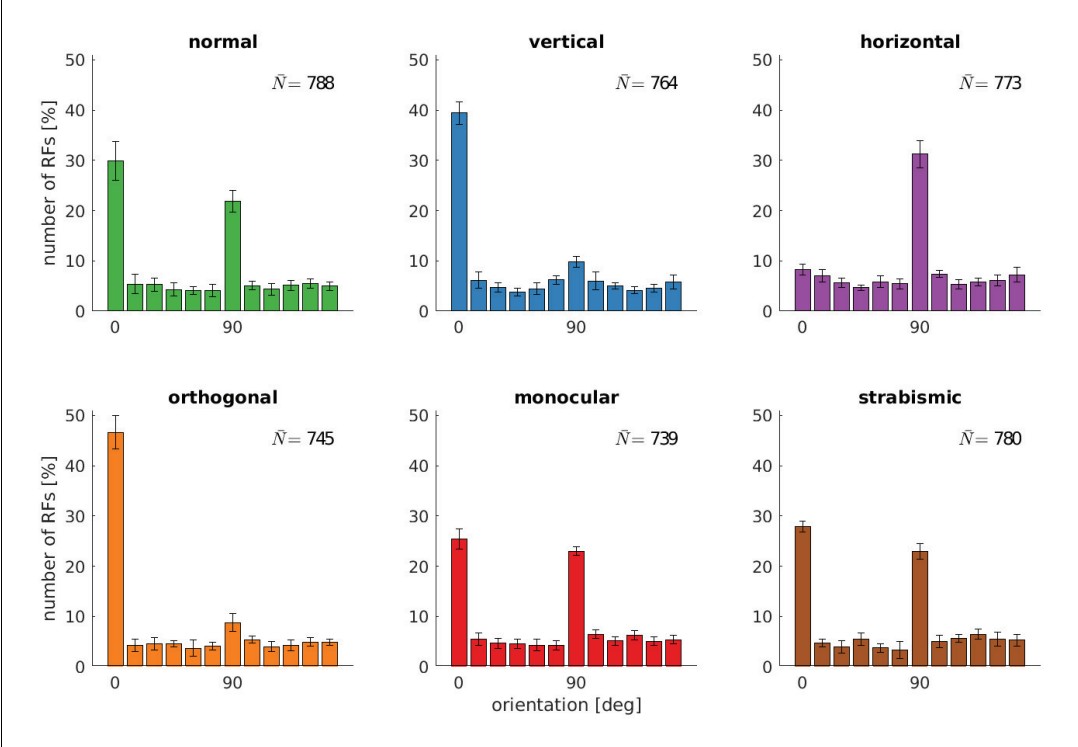

**Figure 3.** Distributions of RFs' orientation preference for different rearing conditions. Displayed are the preferred orientations resulting from fitting Gabor wavelets to the learned RFs of the left eye. Coarse and fine scale RFs have been combined (800 in total). The error bars indicate the standard deviation over five different simulations. $\bar{N}$ describes the average number of RFs that passed the selection criterion for the quality of Gabor fitting (see Materials and methods).

The online version of this article includes the following source data for figure 3:

**Source data 1.** Orientation tuning for all rearing conditions.

orthogonal, monocular, and strabismic case. Most cells become monocular, with a symmetric distribution for orthogonal and strabismic rearing. Monocular deprivation of the right eye leads to a distribution of binocularity indices that is biased toward the left eye.

Comparing our model to biological data, the model's pronounced peak of binocularity indices close to 0 in the normal case matches experimental findings (Figure 1 in *Wiesel and Hubel, 1963* and Figure 5 in *Hubel and Wiesel, 1965*). Animals trained on inputs deprived of certain orientations (Figure 6B in *Stryker et al., 1978*) develop more monocular neurons, but most neurons remain binocular. This is in good agreement with our model.

*Stryker et al., 1978* also reared kittens on orthogonal input and report an increase in monocular neurons (their Figure 6A) when compared to the normal rearing data from Hubel and Wiesel. In comparison to the rearing on horizontal or vertical stripes, there are fewer binocular cells. The loss of binocular neurons that we see in our data is also reported in *Hirsch and Spinelli, 1970*, who reared kittens on orthogonal stripes.

Finally, monocular rearing and the analysis of binocularity was performed in *Wiesel and Hubel, 1963*. In their Figures 3 and 5, we see the development of completely monocular cells after visual deprivation of the other eye. The strabismic case was studied a few years later in *Hubel and Wiesel, 1965* (their Figure 5A) and revealed a division of the neural population in monocular neurons for either left or right eye, in agreement with our model.

## Alternate rearing conditions reduce the number of disparity tuned neurons

A central aspect of the development of binocular vision is the emergence of neurons which are tuned to binocular, horizontal disparities. We therefore investigate how alternate rearing affects the

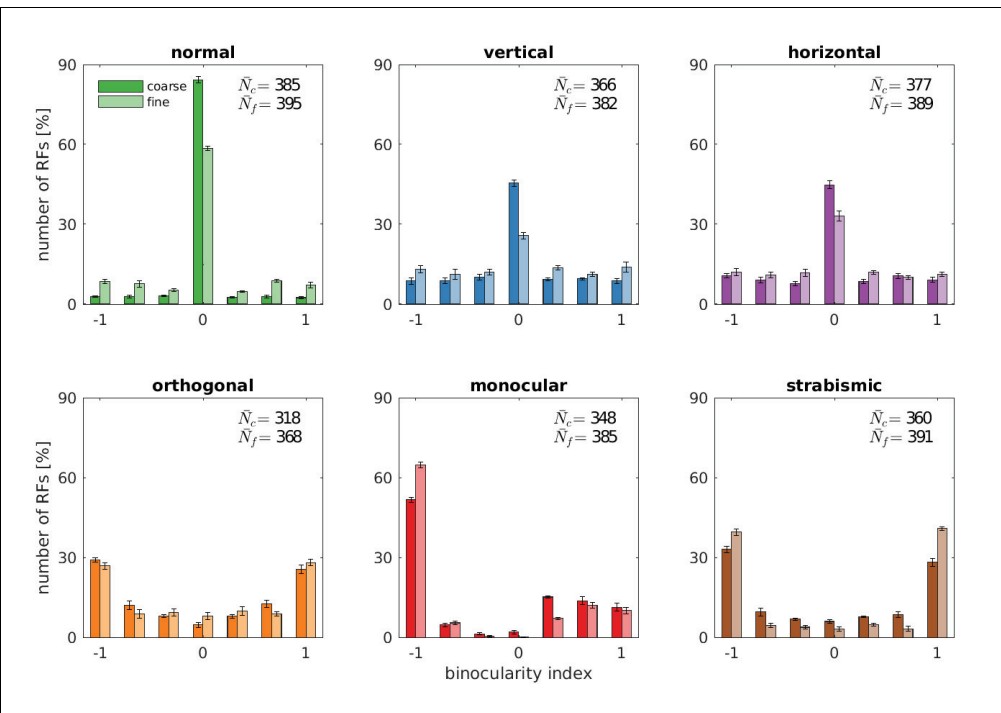

**Figure 4.** Binocularity distributions for different rearing conditions. The binocularity index ranges from −1 (monocular left) over 0 (binocular) to 1 (monocular right). Error bars indicate the standard deviation over five different simulations. $\bar{N}_c$ and $\bar{N}_f$ are the average number of basis functions (out of a total of 400) that pass the selection criterion for Gabor fitting (see Materials and methods).

The online version of this article includes the following source data for figure 4:

**Source data 1.** Binocularity values for all rearing conditions for coarse and fine scale.

number of neurons with disparity tuning and the distribution of their preferred disparities. We estimate horizontal disparity tuning by considering phase shifts between left and right RFs in the following way: We fit binocular Gabor wavelets to the RFs, where all parameters, except for the phase shift, are enforced to be identical for the left and right monocular RF. The disparity for one neuron can then be calculated as described in Materials and methods. The distribution of disparity tuning of the coarse scale neurons is shown in *Figure 5* for the different rearing conditions. Results for the fine scale are comparable and presented in *Figure 5—figure supplement 1*. First, there is a noticeable difference in the number of cells that are disparity tuned between the different rearing conditions: In the normal case, we find the highest number of disparity tuned cells, rearing in a striped environment reduces the number, and uncorrelated input results in the smallest number of disparity tuned cells. In every case, the distribution of preferred disparities is peaked at zero. The height of this peak is reduced for rearing conditions with in-congruent input to the two eyes.

Comparing the normal with the vertical and horizontal case, there is an increase in the number of cells that are tuned to non-zero disparities. This indicates that under these rearing conditions, the agents are exposed to non-zero disparities more often. This is in good agreement with the results from the next section (also see *Figure 6*), where we will see that those models perform less accurate vergence movements compared to the normal case.

In the strabismic case, a neuron's receptive fields in left and right eye are driven by un-corresponding input. This results in very few disparity tuned cells that exhibit a much broader distribution of preferred disparities.

To investigate the effect of a less severe strabismus we conduct an additional experiment similar to *Shlaer, 1971* (see their Figure 2). Here, we fix the strabismic angle to 3°, which results in a corresponding image in the two eyes because one input patch in the coarse scale covers an angle of 6.4°. *Figure 5—figure supplement 2* shows that this leads to an increased amount of disparity tuned cells

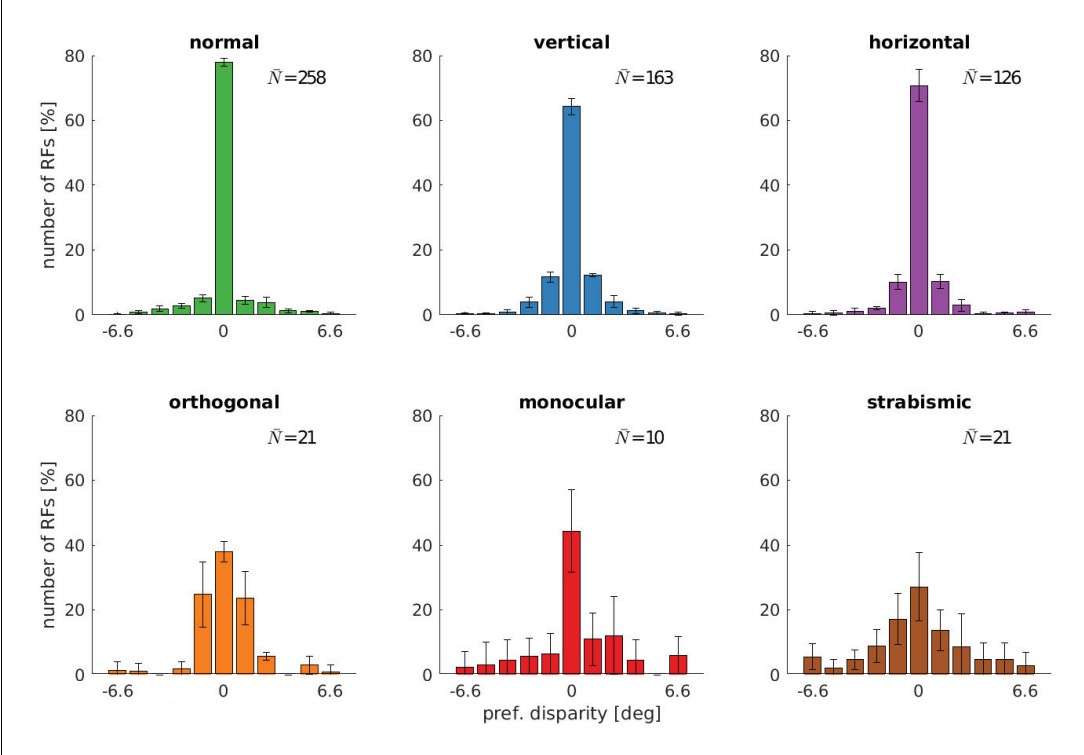

**Figure 5.** Distributions of neurons' preferred disparities for different rearing conditions. The neurons' preferred disparities are extracted from the binocular Gabor fits. Presented are the averaged data for the coarse scale from five simulations. $\bar{N}$ describes the average number of neurons meeting the selection criteria (see Materials and methods).

The online version of this article includes the following source data and figure supplement(s) for figure 5:

**Source data 1.** Coarse scale disparity tuning for all rearing conditions.
**Source data 2.** Fine scale disparity tuning for all rearing conditions. .
**Source data 3.** Coarse scale disparity tuning for training with a constant angle of 3˚.
**Figure supplement 1.** Disparity tuning for the fine scale.
**Figure supplement 2.** Disparity tuning for training with a constant strabismic angle of 3˚.

and a shift of their preferred disparity to 3˚. Exactly as in *Shlaer, 1971*, the constant exposure to a certain disparity leads to a preference for that disparity for the majority of cells.

## Model predicts how alternate rearing conditions affect vergence learning

While the effect of alternate rearing conditions on receptive fields of visual cortical neurons is well studied, there has been little research on the effect of alternate rearing conditions on vergence behavior.

*Figure 6A* shows the evolution of the absolute vergence error, that we interpret as the models' stereo acuity, over the training time for the different rearing conditions. The models with *normal* or *vertical* rearing learn to verge the eyes on the same point on the object, resulting in the reduction of the vergence error to small values of around 0.3˚. The model that learns on images without vertical edges (*horizontal* case) also learns vergence behavior, but does not reach the accuracy of the former models. The *orthogonally*, *monocularly*, and *strabismically* reared models show only random vergence movements and do not improve throughout the training period. Since we use the same random seeds for all simulations, including the initial weights and order of input images, the only difference between these models is the filtering applied to the images (the different rearing conditions). That difference alone is not sufficient to influence the behavior significantly. That is why the results for these three models overlap completely in *Figure 6A*.

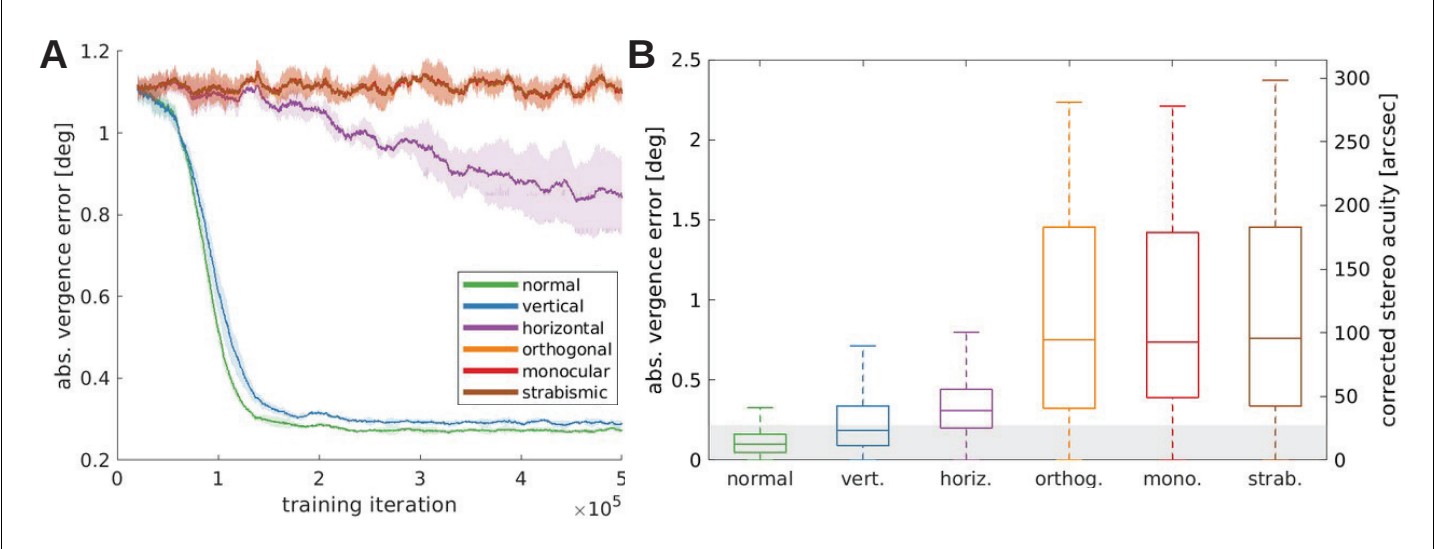

**Figure 6.** Vergence performance of models trained under different rearing conditions. (**A**) Moving average of the vergence error during training. The vergence error is defined as the absolute difference between the actual vergence angle and the vergence angle required to correctly fixate the object. Shaded areas indicate the standard deviation over five different simulations. The curves for *orthogonal*, *monocular,* and *strabismic* conditions are overlapping, see text for details. (**B**) Vergence errors at the end of training after correction of any visual aberrations. Shown is the distribution of vergence errors at the end of a fixation (20 iterations) for previously unseen stimuli. Outliers have been removed. The gray shaded area indicates vergence errors below 0.22°, which corresponds to the model's resolution limit. The second y-axis shows values corrected to match human resolution (see Materials and methods for details).

The online version of this article includes the following source data for figure 6:

**Source data 1.** Training performance for all rearing conditions recorded every 10 iterations.
**Source data 2.** Performance at testing for all rearing conditions.

The main difference to the models that were able to learn vergence is that under these conditions the left and right eye are provided with incongruent input. The orthogonal model receives two monocular images that retain different orientations. The right monocular image of the monocularly deprived model contains little information at all, and the two eyes are physically prevented from looking at the same object in the strabismic case. In these cases, very few neurons with disparity tuning emerge (compare previous section) that could drive accurate vergence eye movements.

## Vision remains impaired if input is corrected after the critical period

In biological vision systems, alterations of the visual input during a critical period of visual development (e.g. due to astigmatism or cataract or experimentally induced alternate rearing conditions) lead to lasting visual deficits that can remain after visual input has been corrected. To test if a similar phenomenon arises in the model, we first train the model with alternate rearing conditions as described above. Then, we freeze all its synaptic weights and study its behavior for *normal* visual input. Specifically, objects are presented at distances $\{0.5, 1, \ldots, 6\}$ m, the initial vergence error is chosen randomly between $-2$ and $2°$, and 40 stimuli that were not seen during training are applied on the object plane. From these initial conditions, we simulate fixations of 20 iterations and record the vergence error at the end.

The results of this testing are displayed in *Figure 6B*. Here, the gray-shaded area indicates a vergence error of 1 pixel. The *normally* trained model exhibits the best performance and actually achieves sub-pixel accuracy in the great majority of trials. The model is more accurate here than in the training phase, because there is no exploration noise during action selection in this testing procedure. Performance declines somewhat for the *vertical/horizontal* models, which were trained on input without horizontal/vertical edges, respectively. Finally, performance for the *orthogonal, monocular* and *strabismic* models is very poor. This is due to their incongruent input to both eyes during training, which impairs the development of binocular neurons tuned to different disparities. Since

this is the first study to investigate the quality of learned vergence movements after exposure to alternate rearing conditions (to the best of our knowledge), the differences in performance are a genuine prediction of our model.

To gain a deeper insight into the underlying mechanisms, we consider the model's *reward land-scape* after training under the different rearing conditions. The model's reward is the *negative reconstruction error* of the sparse coders. This means that vergence angles that result in a low reconstruction error will be preferred. *Figure 7* shows the averaged reconstruction error over three different object distances and ten stimuli for the different rearing conditions. In the normal, vertical, and horizontal case, there is a pronounced minimum at zero disparity, which drives the model to fixate on the same point with both eyes. This is in contrast to the orthogonal, monocular, and strabismic conditions, where the reward landscape is flat, that is, there is no incentive to align the two eyes onto the same point.

## Model captures stereo vision deficits in aniseikonia and predicts increased number of neurons tuned to vertical disparities

Aniseikonia is a condition characterized by a perceived difference in the sizes of left and right eye images. It can occur naturally as result of anatomical or refraction differences of the two eyes, different spacing of photoreceptors in the retinas, or other neurological causes (*South et al., 2019*). Aniseikonia can also be induced as result of the treatment of anisometropia (different refractive powers of the eyes) (*Achiron et al., 1997*). In this scenario, spectacles or artificial lenses are used to correct the refractive power of one or both eyes to create a sharp image in both eyes. However, due to optical magnification this also leads to a difference in the image sizes. When this difference remains small (typically lower than 3%), it can be tolerated by the visual system. Larger values on the other hand lead to problems in fusing the images and a loss of stereopsis (*Katsumi et al., 1986*; *Oguchi and Mashima, 1989*; *Achiron et al., 1997*). Aniseikonia may also occur in anisometropic

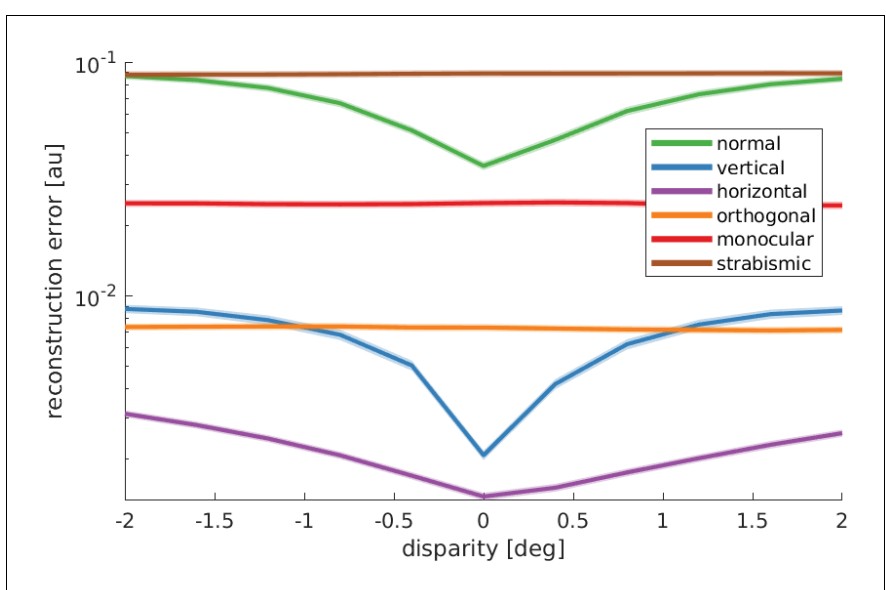

**Figure 7.** Reward landscape at the end of training for the different rearing conditions. Shown is the logarithm of the sparse coder's reconstruction error as a function of disparity. The negative reconstruction error is used as the reward for learning vergence movements. Data are averaged over 10 stimuli not encountered during training, three different object distances (0.5, 3, and 6 m), and five simulations for every condition. The shaded area represents one standard error over the five simulations. Only those models that receive corresponding input to left and right eye display a reconstruction error that is minimal at zero disparity. These are the only models that learn to verge the eyes.

The online version of this article includes the following source data for figure 7:

**Source data 1.** Rewards for 5 random seeds, 3 object distances, 11 different disparity values, 10 different input images and 2 scales for all six rearing conditions.

patients after cataract surgery with implanted intraocular lenses (*Katsumi et al., 1992*; *Gobin et al., 2008*). A recent study reported 7.8% measured aniseikonia in an outpatient clinic cohort (*Furr, 2019*).

Since little is known about the effects of aniseikonia on visual development or the potential benefits of correction (*South et al., 2020*), we conducted a series of experiments to simulate the effects of aniseikonia. We achieve this by simply scaling up the right image by a certain factor and cutting the edges so left and right images retain the same size. The rest of the training procedure remains unchanged.

*Figure 8A* shows the improvement of the stereoscopic acuity as measured by the absolute vergence error as a function of training time for four values of aniseikonia: 0%, 10%, 15% and 25%, where 0% aniseikonia corresponds to the *normal* model from previous sections. Ten percent of aniseikonia leads to slower learning and a slightly reduced vergence performance. While an improvement in vergence performance is still present for 15%, it completely fails for 25%. The increased size of the right image leads to partly incongruent input to the two eyes. As a result, an increased number of monocular RFs develops (*Figure 8—figure supplement 1A*, see *Figure 8—figure supplement 2* for a full set of RFs). The lack of congruent input to both eyes and the resulting lack of binocular receptive fields impairs the development of correct image fusion.

The different object sizes in the left and right image also lead to *vertical disparities*. For example, when fixating the center of a square, the upper edge of the square will be projected to different vertical positions for the two eyes due to the different sizes of the square in the two eyes. We can measure these vertical disparities in a similar way as we measured the horizontal disparities before (see Analysis of receptive fields). *Figure 8B* shows that the number of neurons tuned to vertical disparities initially increases with growing aniseikonia but then reduces again for 25% of aniseikonia. The key to understanding this phenomenon is considering binocularity (*Figure 8—figure supplement 1A*) and vergence performance (*Figure 8A*): at 25% aniseikonia, vergence behavior does not

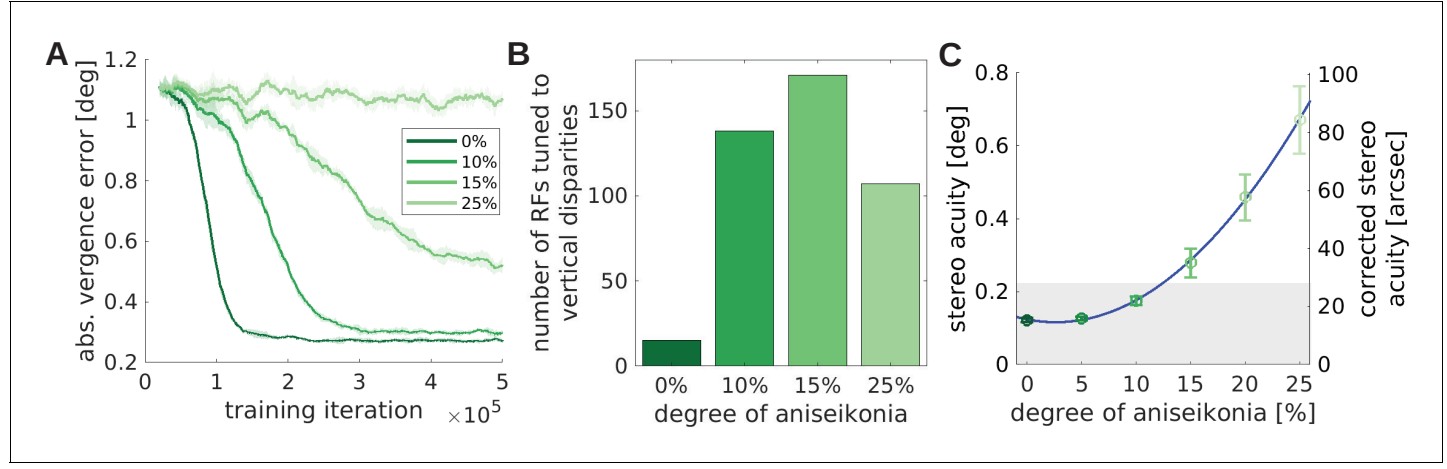

**Figure 8.** The effect of unequal image size (aniseikonia) on the development of binocular vision. (**A**) Vergence error as a function of time for different degrees of aniseikonia. (**B**) Number of RFs tuned to *vertical disparities* for different degrees of aniseikonia during learning. (**C**) Stereo acuity when different degrees of aniseikonia are introduced after normal rearing. The solid line depicts a quadratic fit to the data. The *corrected stereo acuity* on the right y-axis corrects for the lower visual resolution of the model compared to humans.

The online version of this article includes the following source data and figure supplement(s) for figure 8:

**Source data 1.** Vergence error over training time measured every 10 iterations for all degrees of aniseikonia (five random seeds). .

**Source data 2.** Tuning to vertical disparities for all degrees of aniseikonia (five random seeds).

**Source data 3.** Vergence performance of *normal* models tested under different degrees of aniseikonia (five random seeds).

**Source data 4.** Binocularity values for all degrees of aniseikonia.

**Source data 5.** Orientation tuning for all degrees of aniseikonia.

**Source data 6.** Alls RFs for all degrees of aniseikonia.

**Figure supplement 1.** Binocularity, orientation tuning, and the number of RFs tuned to the vertical orientation for models trained under different degrees of aniseikonia.

**Figure supplement 2.** All RFs for different degrees of aniseikonia.

develop and many neurons assume monocular RFs. This reduces the total number of neurons which are tuned to vertical disparites. The inverted U-shaped amount of RFs tuned to vertical disparities for increasing amounts of aniseikonia is a testable prediction of the model.

Interestingly, when looking at the orientation tuning in *Figure 8—figure supplement 1B*, we observe that the number of cells tuned to the *vertical orientation* decreases with increasing aniseikonia (see *Figure 8—figure supplement 1C* for a direct comparison). Since the distribution of orientations in the input images does not change by changing image size, we attribute this change in orientation preference to the model's (in-)ability to perform accurate vergence movements. As we will elaborate in the next section, the ability to detect different ranges of horizontal disparities results in an abundance of vertically tuned cells. When the visual system looses the ability to detect horizontal disparities and to verge, the number of vertical RFs decreases.

We also test the effects of a suddenly induced aniseikonia on a fully developed healthy visual system. *Lovasik and Szymkiw, 1985* induced aniseikonia in healthy subjects and let them perform the Randot and Titmus stereo acuity tests. They found that the stereo acuity diminishes roughly quadratically with the level of aniseikonia. We simulate their experiments by taking a normally trained, healthy model, induce aniseikonia, and test it under the same conditions as before: frozen synaptic weights, novel test stimuli, and a whole range of different object distances and initial vergence errors. We interpret the mean absolute vergence error as the stereo acuity of that model. *Figure 8C* shows that the stereo acuity declines approximately quadratically with increasing aniseikonia as observed by *Lovasik and Szymkiw, 1985*. When we correct for the model's lower visual resolution compared to humans (see Materials and methods), we find that the stereo acuity achieved by the model falls in the typical range of human stereo acuity (*Coutant and Westheimer, 1993*; *Bohr and Read, 2013*). In fact, our model appears to be somewhat more robust against larger values of aniseikonia than human subjects (*Lovasik and Szymkiw, 1985*; *Atchison et al., 2020*). We speculate that this is due to the absence of an interocular suppression mechanism in our model that may accentuate the effects of aniseikonia on stereo vision in humans.

## Model predicts how vergence influences the statistics of orientation preference

Our model also allows us to investigate, for the first time, how the quality of the vergence control influences the neural representation. As a baseline, we consider the orientation tuning of a reference model which is trained on normal visual input and learns an appropriate vergence policy. For simplicity, we only consider the fine scale in the following. We compare this model to a version that is trained on the same input images, but could not verge the eyes. Specifically, the sparse coder is trained normally, but the RL part is disabled. This model sees different disparities during training by looking at objects at different depths, but is not able to change this distribution of disparities to facilitate the encoding. We refer to this model as the 'random disparity' model. In another version of the model, we artificially always set the vergence angle to correctly fixate the objects. In this way, this model is never exposed to non-zero disparities (except for very small ones in the periphery that arise because of slightly different perspectives in the left and right eye). We refer to this version as the 'zero disparity' model.

*Figure 9A* shows the fraction of neurons that are tuned to vertical orientations ($0 \pm 7.5°$) for these three models. When the influence of the RL agent is removed, we observe a significant decrease in the number of vertically tuned neurons. This change must be caused by the different distributions of disparities that the models experience due to their different motor behavior, because all other parameters remain unchanged. In the model that was trained without disparities, we find the least amount of neurons tuned to vertical edges.

To study the role of the distribution of experienced disparities more systematically, we train the sparse coder on different truncated Laplacian distributions of disparities. The distributions are heavy-tailed and centered around zero. The spread in this distribution is determined by $\sigma_L$, the standard deviation. $\sigma_L = 0$ means zero disparity all the time (corresponding to the zero disparity case), while the distribution becomes almost uniform for big values of $\sigma_L$. *Figure 9B* shows how the number of vertically tuned neurons changes in response to different values of $\sigma_L$. We find the smallest number of vertically tuned cells when the disparity is zero throughout the whole training. For very large $\sigma_L$ there are more vertical cells, but not as many as for smaller values which are different from

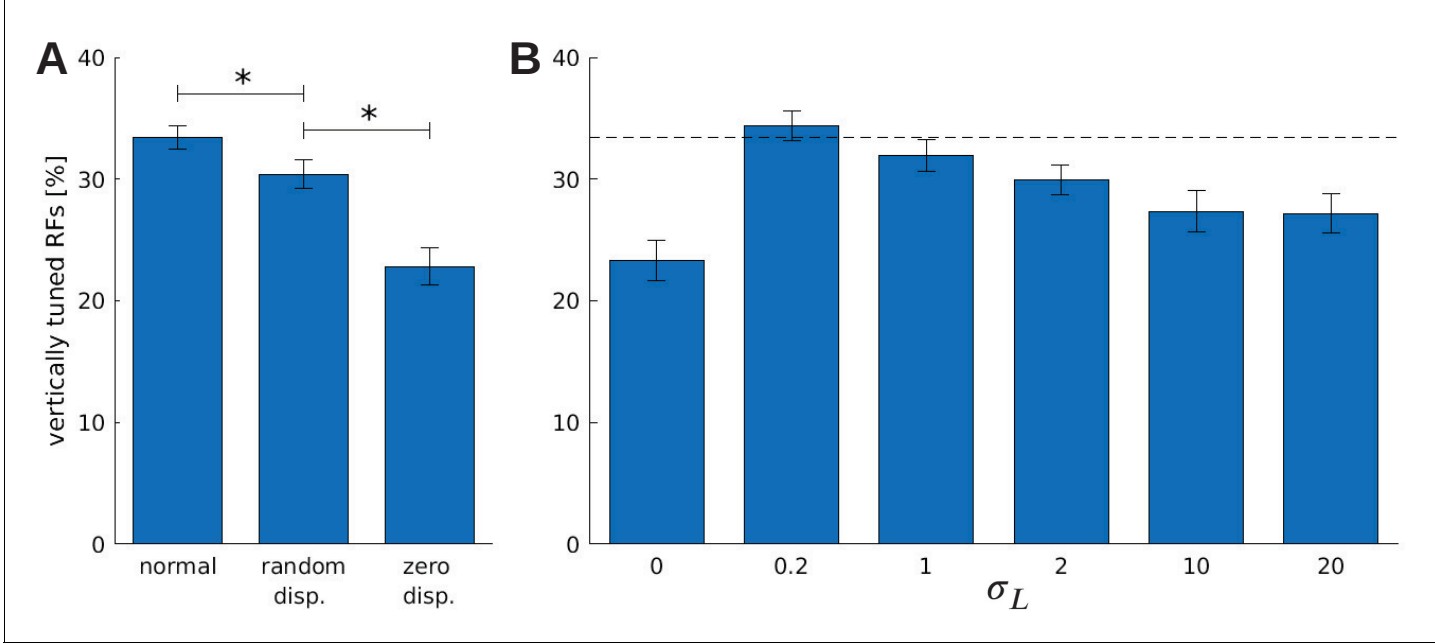

**Figure 9.** The effect of vergence learning on the number of neurons tuned to vertical orientations. (**A**) Fraction of RFs tuned to vertical orientations for different versions of the model (see text for details). Asterisks indicate a statistically significant difference between the samples as revealed by a students t-test (p-values are $7x10^{-3}$ and $1x10^{-3}$). (**B**) Fractions of fine scale RFs tuned to vertical orientations for models trained with (truncated) Laplacian disparity distributions of different standard deviations $\sigma_L$. The value $\sigma_L = 0$ corresponds to 0 disparity all the time, while $\sigma_L = 20$ corresponds to an almost uniform disparity distribution. Error bars indicate the standard deviation over five different simulations. The black dotted line indicates the fraction of vertically tuned RFs in the *normal* model.

The online version of this article includes the following source data and figure supplement(s) for figure 9:

**Source data 1.** Orientation tuning of the three different models. .

**Source data 2.** Orientation tuning of models trained under different Laplacian disparity policies for coarse and fine scale.

**Figure supplement 1.** Number of RFs tuned to vertical orientations for coarse and fine scale combined.

zero. In fact for $\sigma_L = 0.2$, which corresponds to a standard deviation of one pixel in the input image, the number of vertically tuned neurons is maximized.

An intuitive explanation for this over-representation of cells tuned to vertical orientations is given in *Figure 10*. Here, we depict a part of an input image at three different disparities. While the horizontal edge can be encoded by the same RF for all disparity values, the vertical edge demands three different RFs to represent the input pattern faithfully. A system that experiences these disparities in its inputs, needs to devote neural resources to represent them all. If the distribution of disparities becomes too wide, however, individual neurons will receive close to independent input from both eyes and disparities that lie in the range that can be represented by a single RF will be rare. This explains the reduction of the number of vertically tuned RFs for very wide disparity distributions (*Figure 9B*).

## Discussion

A major goal of Computational Neuroscience is the development of models that explain how the tuning properties of neurons develop and how they contribute to the behavior of the organism. Over the last decades, the dominant theoretical framework for understanding the development of tuning properties of sensory neurons has been the *efficient coding hypothesis*. It states that sensory tuning properties adapt to the statistics of the sensory signals. In this framework, the behavior of the organism has been largely neglected, however. Specifically, there has been hardly any work on how developing neural tuning properties shape behavior, how the developing behavior affects the statistics of sensory signals, and how these changing statistics feed back on neural tuning properties. We argue that understanding the development of sensory systems requires understanding this feedback

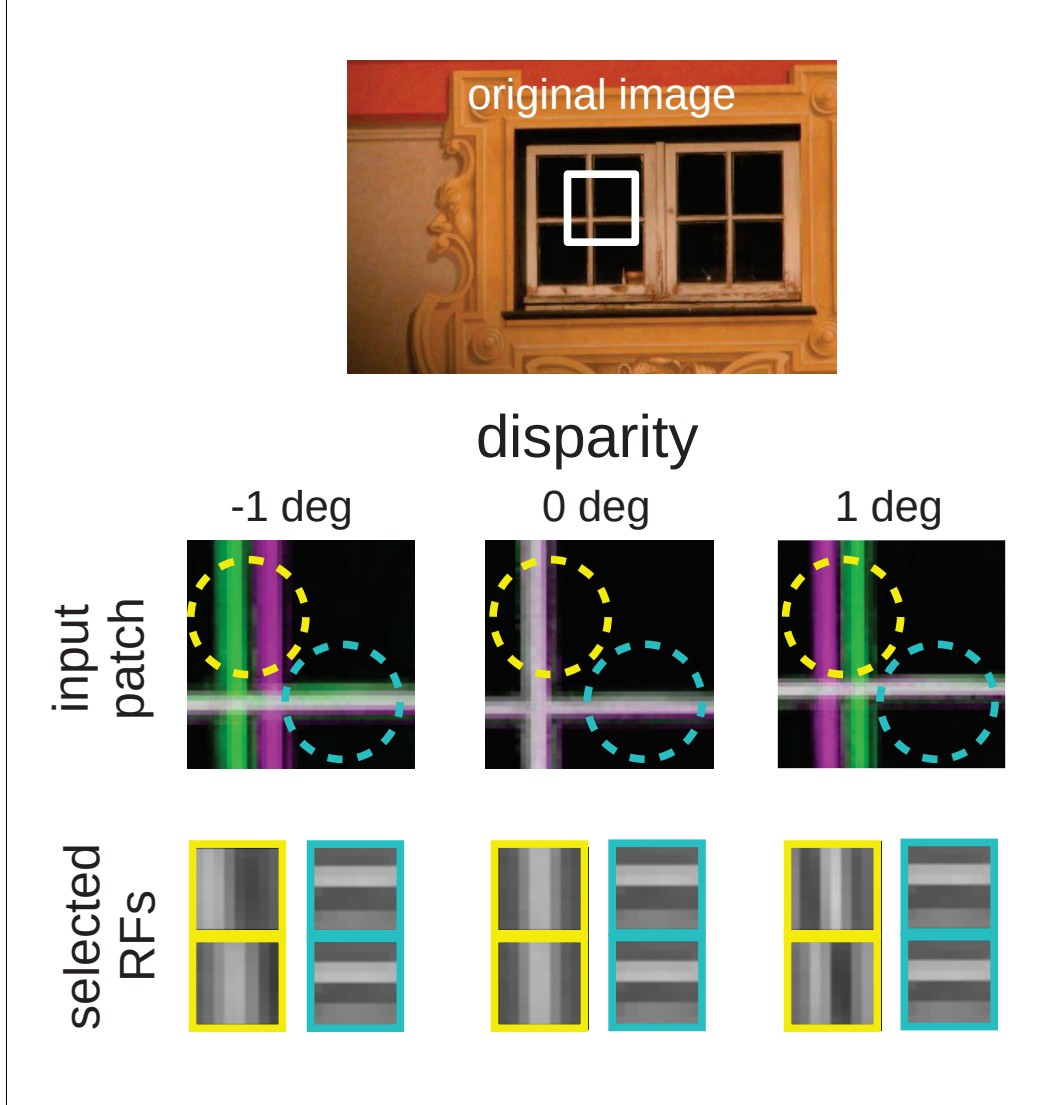

**Figure 10.** Intuition for the over-representation of vertical edges when different disparities have to be encoded. Top: Input scence with marked input patch. Middle: Anaglyph rendering (left: green, right: magenta) of the patch for three different disparities. Two RF locations are highlighted (yellow and cyan circles). Bottom: RFs selected by the sparse coder to encode the inputs. While the RF that encodes the input in the cyan circle is the same for all disparities, the input inside the yellow circle can best be encoded by RFs that are tuned to the corresponding disparities.

cycle between the statistics of sensory signals, neural tuning properties and behavior. The *active efficient coding* (AEC) approach offered here extends classic theories of efficient coding by a behavior component to study this feedback cycle in detail. In AEC, both sensory coding and behavior are adapted to improve the system's coding efficiency. This coupling of perception and action is also a feature of the general framework of *Active Inference* (*Adams et al., 2013*; *Parr and Friston, 2018*). There, motor commands are generated to fulfill sensory predictions, while AEC offers a mechanism to adapt motor commands to improve sensory coding efficiency via reinforcement learning. Interestingly, the presented model does not make use of any efference copy of motor commands to help predict the next sensory input. Evidently, such feedback is not required to learn accurate stereoscopic vision. However, extending our model to incorporate efference copies of motor commands may still be interesting, for example, for the case of calibrating pursuit eye movements, and is left for future work. Interestingly, both AEC and Active Inference have also been linked to higher level cognitive phenomena such as imitation (*Friston and Frith, 2015*; *Triesch, 2013*).

In the present study, we have focused on active binocular vision, where a simulated agent autonomously learns to fixate a target object with both eyes via vergence eye movements. All parts of our model self-organize in tandem to optimize overall coding efficiency. We have identified three critical tests that a model of the development of stereoscopic vision should pass and we have demonstrated that the proposed model passes all of them. Specifically, we have shown that (1) our model autonomously self-calibrates, reaching human-like vergence performance when correcting for differences in visual resolution. Second, it handles random dot stereograms, despite having never been exposed to such stimuli. Third, the model reproduces a wide range of findings from animal studies on alternate rearing conditions, which often show dramatic effects on neural representations and behavior. Beyond explaining the experimental findings, our model also predicts systematic changes in the learned vergence behavior in response to altered rearing conditions. In addition, the model predicts that the learning of accurate vergence behavior systematically influences the neural representation and offers a novel explanation for why vertical orientations tend to be over-represented in visual cortex compared to horizontal ones, at least in primates (*De Valois et al., 1982b*) and humans (*Yacoub et al., 2008*; *Sun et al., 2013*). These predictions should be tested in future experiments. For example, in unilaterally enucleated animals, a bias in favor of vertical orientations over horizontal ones may be reduced or completely absent (*Frégnac et al., 1981*).

By freezing the neural network after the training period, we also simulated the state of the brain after the critical period. Even after correcting the optical aberrations present during training we observed a reduced vergence performance for all alternate rearing regimes. This finding is in line with a large body of evidence suggesting that optical aberrations should be corrected as early as possible to facilitate healthy development of binocular vision (e.g. *Daw, 1998*; *Fawcett et al., 2005*, but also see *Ding and Levi, 2011*).

While our results qualitatively match experimental findings, there are some quantitative differences. In particular, while the distribution of binocularity indices (*Wiesel and Hubel, 1963*) and disparities (*Sprague et al., 2015*) in healthy animals are relatively broad (*De Valois et al., 1982a*; *Stevenson et al., 1992*; *Ringach et al., 1997*), we find more narrow ones in our model. These differences are likely due to a number of simplifications present in our model. In the brain, inputs from both eyes into primary visual cortex are organized into ocular dominance bands such that individual cortical neurons may receive input which is already biased toward one or the other eye (*LeVay et al., 1980*; *Crowley and Katz, 2000*). In contrast, in our model all neurons receive similar amounts of input from both eyes and are therefore already predisposed for becoming binocular cells. This might explain the model's narrower distribution of binocularity indices. Regarding the distribution of preferred disparities, animals raised under natural conditions will experience a broad range of disparities in different parts of the visual field, since objects in different locations will be at different distances. In the model, the visual input is quite impoverished, as it is dominated by a single large frontoparallel textured plane. Once this plane is accurately fixated, most parts of the visual field will appear at close to zero disparity. This may explain the narrower distribution of preferred disparities observed in the model.

Similarly, the distribution of preferred orientations in our model shows a very strong preference for horizontal and vertical, that is accentuated in comparison to biological data (*Li et al., 2003*; *De Valois et al., 1982b*). Possible reasons for this include the discrete, rectangular pixel grid with which visual inputs are sampled, the choice of our image data base (*Olmos and Kingdom, 2004a*), which contains mostly man-made structures including buildings, etc., for which it is known that they contain an abundance of horizontal and vertical edges (*Coppola et al., 1998*), and the model's restriction to the central portion of the visual field, where the oblique effect is more pronounced (*Rothkopf et al., 2009*). To clarify the role of the input images, we repeated the main findings with a random selection of all sections from the McGill Database (Appendix 1) and indeed found that the over-representation of vertical and horizontal orientations is reduced.

Next to addressing the above limitations, an interesting topic for future work is to use the model to study the development of amblyopia. For this, we have recently incorporated an interocular suppression mechanism, since suppression is considered a central mechanism in the development of amblyopia (*Eckmann et al., 2019*). Such models could be a useful tool for predicting the effectiveness of novel treatment methods (*Papageorgiou et al., 2019*; *Gopal et al., 2019*).

In conclusion, we have presented a computational model that sheds new light on the central role of behavior in the development of binocular vision. The model highlights how stimulus statistics,

sensory representation and behavior are all inter-dependent and influence one another and how alternate rearing conditions affect every aspect of this system. The Active Efficient Coding approach pursued here may be suitable for studying various other sensory modalities across species.

## Materials and methods

In the following, we describe the different components of the model, the experimental setup, and the analysis techniques. The implementation is publicly available at https://github.com/Klimmasch/AEC/ (copy archived at swh:1:rev:96e9ae2336937469a8f1602c178ea5e0cb8564b6; *Klimmasch, 2021*).

### Image processing

We use OpenEyeSim (*Priamikov and Triesch, 2014*; *Priamikov et al., 2016*) to render the left and right eye image. It comprises a detailed biomechanical model of the human oculomotor system and simulates a 3-dimensional environment. A rectangular plane is moved in front of the learning agent (perpendicular to the gaze direction). On it we apply greyscale textures from the McGill Calibrated Color Image Database (*Olmos and Kingdom, 2004b*) to simulate objects at different depths. Specifically, we chose the man-made section from the McGill Database (*Olmos and Kingdom, 2004a*), because its statistics may resemble the statistics of the indoor environments that a majority of infants grow up in. As a comparison, we repeated our main analysis with a random set of images across all sections of the McGill data base (see Appendix 1). Behind the textured plane there is a large background image, simulating a natural background behind objects of interest. This background image also prevents the agent from receiving trivial input.

Even tough it is possible to place three-dimensional objects inside OpenEyeSim, we opted for rendering natural input stimuli on a flat plane at different depths. On the one hand, this ensures natural input statistics, and on the other hand it enables us to uniquely define the correct vergence angle and measure the model's vergence performance (see Measuring the vergence error).

The two monocular images rendered by OpenEyeSim cover a vertical field of view of 50° and have 320 px × 240 px (focal length $F = 257.34 \, \text{px}$). We use Matlab to extract single patches in different resolutions and combine corresponding patches from the left and right image. These binocular patches will be jointly encoded by the sparse coder. The *coarse scale* corresponds to 128 px × 128 px in the original image (corresponds to 26.6° × 26.6°) and is down-sampled by a factor of 4 to 32 px × 32 px. The *fine scale* image corresponds to 40 px × 40 px (8.3° × 8.3°) and is not down-sampled. From coarse and fine scale we extract 8 px × 8 px patches with a stride of 4 px and combine corresponding left and right patches to 16 px × 8 px binocular patches (see *Figure 1*). One patch in the coarse scale covers a visual angle of 6.6° and in the fine scale one patch covers 1.6°. In total, we generate 81 fine scale and 49 coarse scale patches that are subsequently normalized to have zero mean and unit norm.

### Sparse coding

The patches from coarse and fine scale are used in the sparse coding step to construct a neural representation of the visual input and to generate a reward signal that indicates the efficiency of this encoding. Each scale $S \in \{c, f\}$ comprises a dictionary of binocular basis functions (BFs) $\phi_{S,i} \in \mathcal{B}_S$. We refer to them as receptive fields (RFs) for simplicity. We choose $|\mathcal{B}_s| = 400$ because less would result in a decline in vergence performance and more are computationally more expensive and do not improve performance (*Lelais et al., 2019*).

Each input patch $p_{S,j}$ is reconstructed by a sparse linear combination of 10 BFs:

$$\hat{p}_{S,j} = \sum_{i=1}^{|\mathcal{B}_S|} \kappa_{S,i}^j \phi_{S,i} \,, \tag{1}$$

where the vector of activations $\kappa_S^j$ is allowed to have only 10 non-zero entries. The $\kappa_S^j$ are chosen by *matching pursuit* (*Mallat and Zhang, 1993*). This greedy algorithm selects the 10 BF from the respective dictionary that yield the best approximation $\hat{p}_{S,j}$ of a patch and was chosen for computational efficiency (*Zhang et al., 2015*). Using 10 BFs to encode the input leads to a qualitatively good reconstruction (*Lelais et al., 2019*) and more would be computationally more expensive.

The total reconstruction error $E_S$, where $S \in \{c, f\}$ indicates the scale, is calculated as the sum over all squared differences between all patches and their approximations:

$$E_S = \sum_{j=1}^{|p_S|} ||p_{S,j} - \hat{p}_{S,j}||^2 \, . \tag{2}$$

The total reconstruction errors combined from both scales, $E = E_c + E_f$, is used as the *negative reward* during reinforcement learning (see following section). The average reconstruction errors per patch for each scale are also used to update the BFs via *Gradient descent*. This adaptation is achieved by a simple Hebbian learning rule (*Olshausen and Field, 1996*; *Zhao et al., 2012*):

$$\Delta \phi_{S,i} = \frac{\eta}{|p_S|} \sum_{j=1}^{|p_S|} \kappa_{S,i}^j (p_{S,j} - \hat{p}_{S,j}). \tag{3}$$

This formula implements a simple form of activity-dependent learning between a population of encoding neurons $\kappa_S$ and an error-detection population. $\eta$ is the sparse coder's learning rate and set to 0.2 throughout our simulations when learning was active. Varying this parameter (while $\eta > 0$) just influences the convergence speed of the RFs but does not influence tuning properties. After each update with *Equation 3* the weight vector of a RF is divided by its L2-norm to normalize it to unit length.

In the beginning of training, analogous to the state just before eye opening (*Huberman et al., 2008*; *Hagihara et al., 2015*), we initialize the RFs with random Gabor functions. Specifically, both the left eye and the right eye component of a binocular basis function have the shape of a Gabor function, but the two Gabor functions have independently drawn random orientations. We have verified that the results can also be achieved when RFs are initialized as Gaussian white noise. The use of random Gabors makes the vergence learning more stable and is biologically more plausible (*Albert et al., 2008*).

For the next step (reinforcement learning), we generate a state representation in the form of a feature vector, where every entry describes the mean squared activation of one BF over the whole input image:

$$F_{S,i} = \sum_{j=1}^{|p_S|} \frac{(\kappa_{S,i}^j)^2}{|p_S|} \, . \tag{4}$$

Taken together, this feature vector $F$ has $2|\mathcal{B}_S|$ entries for both scales combined.

With this pooling procedure we simulate the activity of complex cells that integrate the firing rates of multiple simple cells that are distributed over the whole visual space (*Freeman and Ohzawa, 1990*). In that sense, we achieve a marginalization over all positions and estimate what disparities are present in the input image. This is in line with approaches that utilize feature histograms to extract position-invariant features, for example to classify objects (*Swain and Ballard, 1991*; *Mel, 1997*). In these studies, it is common to normalize the coefficients/features in the histograms to make up for different sampling rates, different lighting conditions, etc. We do not need to normalize the pooled values, because in our case, there is a fixed number of active features (10) per image patch.

## Generation of motor commands

The angular position of the eyes are controlled by two extra-ocular eye muscles responsible for rotations around the vertical axis. This *medial* and *lateral rectus* are simulated utilizing an elaborate muscle model (*Umberger et al., 2003*) inside OpenEyeSim (*Priamikov and Triesch, 2014*; *Priamikov et al., 2016*). Since we are interested in vergence movements only, we assume symmetrical eye movements so that the activities of the two muscles are mirrored for both eyes.

In contrast to recent models of *active inference* where a prediction of proprioceptive feedback is send to the muscles (*Adams et al., 2013*; *Parr and Friston, 2018*), we simply add a differential change in muscle innervation to the current muscle innervation. To generate those innervations (between [0, 1] in arbitrary units), we use reinforcement learning (*Sutton and Barto, 1998*). Specifically, the model employs the CACLA+VAR algorithm from *Van Hasselt and Wiering, 2007* that generates outputs in continuous action space. In short, it uses an actor-critic architecture

(*Grondman et al., 2012*), where the actor and critic use neural networks as function approximators. These neural networks receive the state vector $s_t$ that is the concatenation of the BF activations from both scales (see previous section) and the current muscle innervations. The entries in $s_t$ are scaled by Welford's algorithm (*Welford, 1962*) to have zero mean and a fixed standard deviation.

The critic is a one-layer network that aims to learn the value of a state. From the state vector it approximates the discounted sum of all future rewards

$$V(s_t) = \sum_{i=0}^{\infty} \gamma^i r_{t+i} , \tag{5}$$

where $r_t$ represents the reward achieved at time $t$ and $\gamma$ is the discount factor. To update this value network, we calculate the *Temporal Difference Error* (*Tesauro, 1995*; *Sutton and Barto, 1998*) as $\delta_t = r_t + \gamma V_t(s_{t+1}) - V_t(s_t)$. The parameters of the critic, $\theta^V$, are initialized randomly and updated by

$$\Delta\theta_{i,t}^V = \alpha \delta_t \frac{\partial V_t(s_t)}{\partial \theta_{i,t}^V}, \tag{6}$$

where $\alpha$ represents the learning rate for updating the critic.

The actor is an artificial neural network with one hidden layer with $\tanh$ activation functions and a two-dimensional output that depicts changes in muscle innervation for the two relevant eye muscles (lateral and medial rectus). The generated motor outputs are random in the beginning and the network is updated whenever the given reward was higher than estimated by the critic:

$$\text{IF} \quad \delta_t > 0 : \Delta\theta_{i,t}^A = \beta(a_t - A_t(s_t))\frac{\partial A_t(s_t)}{\partial \theta_{i,t}^A}\left[\frac{\delta_t}{\sqrt{\text{var}_t}}\right], \tag{7}$$

where $\beta$ is the actor's learning rate, $A_t(s_t)$ is the action selected by the actor at time $t$, and $a_t = A_t(s_t) + \mathcal{N}(0, \sigma^2)$ is the action that is actually executed. Adding Gaussian noise to the actor's output to discover more favorable actions is called *Gaussian exploration*. The last term scales the update depending on how much better the action was than expected with respect to its standard deviation.

To keep the actor's weights in check, we use a weight regularizer $g$ in a scaling operation:

$$\theta_{i,t}^A \leftarrow \theta_{i,t}^A(1 - (g\beta)) . \tag{8}$$

The convergence of the RL algorithm (*Van Hasselt and Wiering, 2007*) is only guaranteed when the critic learns to represent the reward landscape on a faster timescale than the actor learns to find appropriate actions. Including this constraint, we conducted an exhaustive grid-based search for parameters that would minimize the *root mean square error* (RMSE) of the *vergence error* (see Measuring the vergence error) after 0.5 million iterations while ensuring the median being close to 0. The critic learning rate $\alpha$, the actor learning rate $\beta$, and the discount factor $\gamma$ were varied between 0 and 1. The results were more stable when $\beta$ decayed to 0. The number of hidden units in the actor $L$ was varied between 5 and 500, explorative noise $\sigma^2$ and weight regularization $g$ between $10^{-4}$ and $10^{-6}$, and the standard deviation in the feature vector $std_{feature}$ between $10^{-1}$ and $10^{-3}$. The following parameters were found to be optimal and are used throughout all experiments in this paper: $\alpha = 0.75$, $\beta$ starts at 0.5 and linearly decreases to 0, $\gamma = 0.3$, $L = 50$, $\sigma^2 = 10^{-5}$, $g = 10^{-5}$, and $std_{feature} = 2x10^{-2}$.

## Simulation of alternate rearing conditions

The deprivation of oriented edges is simulated by convolving the input images with elongated Gaussian kernels defined by:

$$K_{\sigma_x,\sigma_y}(x,y) = \exp\left(-\left(\frac{x^2}{2\sigma_x^2} + \frac{y^2}{2\sigma_y^2}\right)\right) , \tag{9}$$

where $\sigma_{x/y}$ represent the standard deviation in the horizontal/vertical direction.

Kernels with a large $\sigma_x$ ($\sigma_y$) will blur out vertical (horizontal) edges. Specifically, to simulate the deprivation of horizontal orientations, $\sigma_x$ is set to 33 px (to cover one patch in the coarse scale completely) and $\sigma_y$ to a small value of 0.1 px. The numbers are reversed for the deprivation of vertical orientations. In the case of orthogonal rearing, the left eye receives an image deprived of horizontal orientations while the right eye receives one without vertical orientations. To make up for the small standard deviation of 0.1 in the direction that should not be impaired, the images in the *normal* case are convolved with a Gaussian kernel with $\sigma_x = \sigma_y = 0.1\,\mathrm{px}$. The resulting filters are displayed in *Figure 11*.

To simulate monocular deprivation (MD) we set $\sigma_x = \sigma_y = 240\,\mathrm{px}$ for the right input image only. The small patches that we extract from this strongly blurred image contain almost no high spatial frequencies.

A strabismus is artificially induced by rotating the right eye ball inwards as it is commonly done in biological experiments by fixating a prism in front of the eye or by cutting part of the lateral rectus muscle. In our Open-Eye-Simulator, however, we can rotate the eye by a specific angle. One input patch in the coarse scale covers 6.6°. When we set the strabismic angle to 3° there is still an overlap in the input images that will be reflected in the neural code. In contrast, when the strabismic angle is set to 10°, the input patches become completely uncorrelated. Examples of the changes done to the input images are displayed in *Figure 2*.

## Analysis of receptive fields
### Gabor fitting
To determine the orientations of the RFs we use MATLAB's implementation of the *trust region reflective algorithm* for non-linear curve fitting (*Coleman and Li, 1996*) to fit them to two-dimensional Gabor functions as defined by:

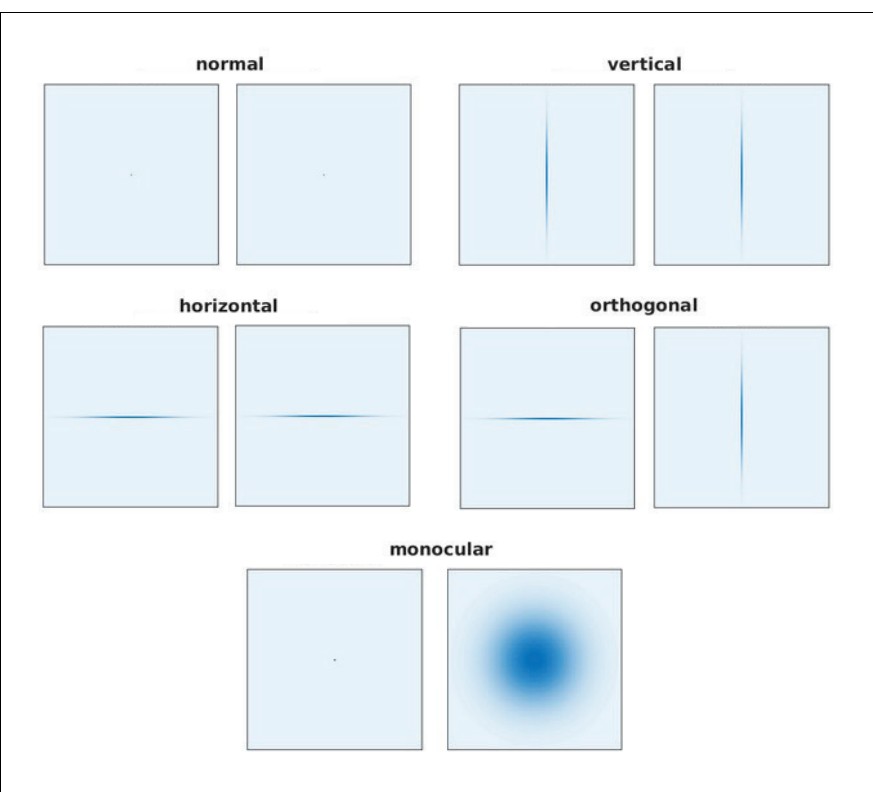

**Figure 11.** The filters used for the normal, vertical, horizontal, othogonal, and monocular models.
The online version of this article includes the following source data for figure 11:

**Source data 1.** The actual filters used during training.

$$G(x,y,\theta,f,\psi,\sigma,\xi) = \exp\left(-\frac{x'^2 + \xi^2 y'^2}{2\sigma^2}\right)\cos(2\pi f x' + \psi), \tag{10}$$

with $x' = x\cos(\theta) + y\sin(\theta)$ and $y' = -x\sin(\theta) + y\cos(\theta)$.

Here, $f$ denotes the frequency, $\psi$ the phase offset, $\sigma$ the standard deviation of the Gaussian envelope, $\xi$ the spatial aspect ratio and $\theta$ the orientation, where $\theta = 0°$ corresponds to a vertically oriented Gabor function. We initialize the parameters randomly 150 times and fit the function either to the left or right RFs (or to both, see below). To evaluate the quality of the fits, we record the difference between the actual RFs and the Gabor fit. More specifically, the *residual* is defined as the sum of the squared differences in single pixel values between RFs and the fit. To compare the fits across the different experimental conditions, we only took those fits where this residual was less than or equal to 0.2. This accounts for more than 96% of all RFs in the healthy case. Another interpretation for these fits is a stimulus that maximally activates the particular neuron.

## Binocularity index

To assess the extent to wich a neuron is responsive to inputs from one vs. the other eye, *Hubel and Wiesel, 1962* introduced the binocularity index. They determined a stimulus that maximizes the monocular response, and applied this stimulus separately in left or right eye to get the (monocular) neural responses $L$ and $R$. Hubel and Wiesel then compared the responses and sorted each cell into one of 7 bins. The first bin contained all cells responsive only to the contralateral eye, the 7th contained all cells responsive only to the ipsilateral eye, the 4th contained all binocular cells and the rest was distributed between the other bins. To investigate the binocularity of a cell in the model, we compare their monocular response to the left and right Gabor fit. The eye with the greater response is the dominant eye for this neuron. Similar as in *Hubel and Wiesel, 1962* we show the best stimulus (here the Gabor fit) to the dominant eye and the same stimulus to the non-dominant eye and record the responses $L$ and $R$. We then calculate the binocularity index $b$ as:

$$b = \frac{R - L}{R + L}, \tag{11}$$

such that the resulting binocularity index lies between $-1$ (monocular left) and $+1$ (monocular right), and 0 indicates a perfectly binocular neuron.

## Disparity tuning

To establish the (horizontal) disparity tuning of a binocular model neuron, we fit coupled Gabor functions to the left and right receptive fields. In doing so, we assume that all parameters are equal for the left and right monocular sub-region of the RFs except for the phase offset $\psi$, that can be different for left and right eye. Following the assumption that the disparity tuning in a binocular cell is encoded by means of this phase shift, we can calculate the preferred (horizontal) disparity $d$ of a neuron by:

$$d = \frac{\psi_L - \psi_R}{2\pi f \cos\theta}. \tag{12}$$

The maximally detectable disparity is given by the RF size, that is, the visual angle one binocular patch covers. RFs with a disparity preference bigger than the RF size are excluded from the analysis.

For calculating the preferred *vertical* disparity, we adapt *Equation 12* in the following way:

$$d^{\text{vert}} = \frac{\psi_L - \psi_R}{2\pi f \sin\theta}. \tag{13}$$

For the number of neurons tuned to vertical disparites in *Figure 8B*, we consider only neurons with horizontal orientation preference ($90° \pm 7.5°$) and simply count all neurons in the population that do not fall into the zero disparity bin ($0° \pm 0.6°$).

## Measuring the vergence error

Given the exact distance to an object ($d_o$) and the inter-pupillary distance ($d_I = 5.6\,\text{cm}$) we can calculate the vergence angle desired for perfectly fixating this object as:

$$z^{des} = 2\arctan\left(\frac{d_I}{2d_o}\right). \tag{14}$$

The absolute difference between this angle and the actual angle between the eyes, $z$, is called the vergence error and is used in our experiments to track the model's ability to use active binocular vision:

$$e^{verg} = |z^{des} - z|. \tag{15}$$

In our experiments, we use a textured plane with varying distances instead of a 3D environment. This provides us with an unambiguous measure of the distance to the objects and we can easily calculate $z^{des}$ and $e^{verg}$. While the vergence error can be defined at every time step, we only analyze it at the end of a fixation (corresponding to the last of 10 time steps), to give the model sufficient time to fixate the object.

When we look at the influence of the vergence movements on the neural representation (*Figure 9B*), we artificially set the vergence angle to simulate different disparity distributions. We use Laplacian distributions, centered around 0, with different standard deviations.

The probability density distribution of a Laplacian distributed random variable $X$ is defined as

$$p(x) = \frac{1}{2b}e^{-\frac{|x-\mu|}{b}}, \quad -\infty < x < \infty, \tag{16}$$

where $b = \frac{\sigma_L}{\sqrt{2}}$ is the scaling parameter. We limit the vergence angle to lie between 0 (looking parallel) and 11.4 (looking at 0.28 m). To simulate the disparity distribution, we set $\mu$ to the angle that is desired to fixate an object at a certain distance $d_o$

$$\mu = 2\arctan\left(\frac{d_I}{2d_o}\right) \tag{17}$$

and draw from the distribution. The data shown in *Figure 9B* depict the fine scale only. The results from the two-scale model can be found in *Figure 9—figure supplement 1*.

## Correcting for lower visual resolution of the model compared to humans

Visual resolution in humans is (amongst other factors) constrained by the distance of photoreceptors on the central retina, which is around 2.5 μm (*Curcio et al., 1990*). Translated to visual angle, this corresponds to a resolution of $r_{\text{human}} \approx 28\,\text{arc sec}$ (*Kalloniatis and Luu, 2007*). In the model, visual resolution is constrained by the discrete sampling of the pixel array. Given the focal length $F = 257.34\,\text{px}$ from above, the angular resolution corresponds to $r_{\text{model}} = \arctan\left(\frac{1\,\text{px}}{F}\right) = 0.22° = 802\,\text{arc sec}$. To convert measurements of the model's stereo acuity or vergence accuracy to human values, we therefore apply a conversion factor of $s_{\text{conversion}} = \frac{r_{\text{human}}}{r_{\text{model}}} = 0.035$ to both kinds of values. Note that doing this for vergence accuracy assumes that vergence performance is ultimately limited by visual processing constraints rather than motor constraints. Since our model neglects any motor noise and uses a continuous action space, this assumption is reasonable. We therefore equate vergence error with stereo acuity in the model.

## Acknowledgements

We are grateful to Alexander Lichtenstein for providing access to the OpenEyeSim simulation environment. We also thank the NEURO-Dream consortium for stimulating discussions that inspired part of this work.

## Additional information

### Funding

| Funder | Grant reference number | Author |
|---|---|---|
| Federal Ministry of Education and Research | 01GQ1414 | Lukas Klimmasch<br>Alexander Lelais |
| Federal Ministry of Education and Research | 01EW1603A | Lukas Klimmasch<br>Johann Schneider<br>Jochen Triesch |
| H2020 European Research Council | 713010 | Alexander Lelais |
| Research Grants Council, University Grants Committee | 16244416 | Bertram Emil Shi |
| Quandt Foundation | | Jochen Triesch |
| Federal Ministry of Education and Research | 01EW1603B | Maria Fronius |

The funders had no role in study design, data collection and interpretation, or the decision to submit the work for publication.

### Author contributions

Lukas Klimmasch, Conceptualization, Resources, Data curation, Software, Formal analysis, Validation, Investigation, Visualization, Methodology, Writing - original draft, Writing - review and editing; Johann Schneider, Software, Formal analysis, Methodology, Writing - review and editing; Alexander Lelais, Software, Investigation, Writing - review and editing; Maria Fronius, Conceptualization, Supervision, Writing - review and editing; Bertram Emil Shi, Conceptualization, Investigation; Jochen Triesch, Conceptualization, Supervision, Funding acquisition, Investigation, Methodology, Writing - original draft, Project administration, Writing - review and editing

### Author ORCIDs

Lukas Klimmasch ![iD] https://orcid.org/0000-0002-9923-3052
Jochen Triesch ![iD] http://orcid.org/0000-0001-8166-2441

### Decision letter and Author response

Decision letter https://doi.org/10.7554/eLife.56212.sa1
Author response https://doi.org/10.7554/eLife.56212.sa2

## Additional files

### Supplementary files

• Transparent reporting form

### Data availability

All data generated or analysed during this study are included in the manuscript and supporting files. Source data files have been provided for all Figures displaying our own generated data.

The following previously published dataset was used:

| Author(s) | Year | Dataset title | Dataset URL | Database and Identifier |
|---|---|---|---|---|
| Olmos A, Kingdom FAA | 2004 | McGill Calibrated Colour Image Database | http://tabby.vision.mcgill.ca/ | McGill, tabby.vision.mcgill.ca/ |

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

## Appendix 1

### Influence of the choice of input images

Here we compare our main results that were achieved using the *man-made* section of the McGill database (*Olmos and Kingdom, 2004a*) to those obtained with input images from a random collection of images from this database that also contains the sections *animals*, *foliage*, *flowers*, *fruits*, *landscapes*, *winter*, and *shadows*.

Overall the results are very similar, except the reduction of RFs tuned to the vertical and horizontal orientation. This also results in a reduced effect of changing $\sigma_L$ in *Figure 5*. Since man-made environments typically contain many vertical and horizontal structures it is not surprising that this feature is accentuated in the RFs' statistics as compared to those trained on a random sample of images.

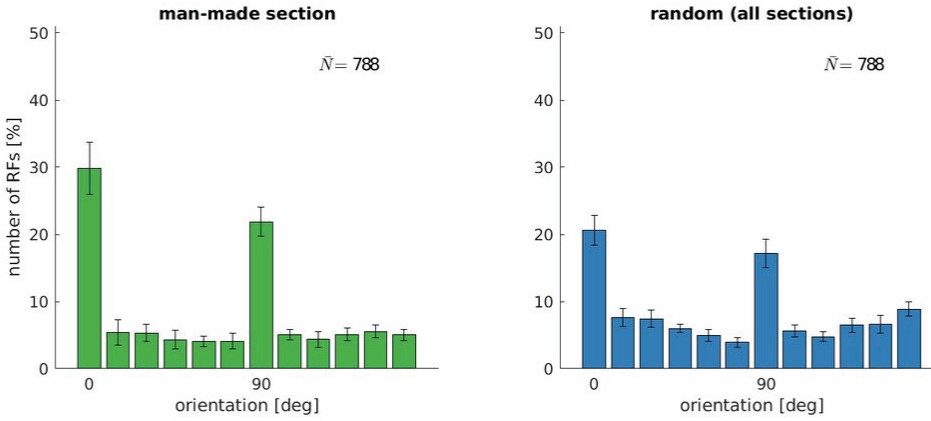

**Appendix 1—figure 1.** Orientation tuning for five models trained with the man-made section or a random sample from the McGill database.

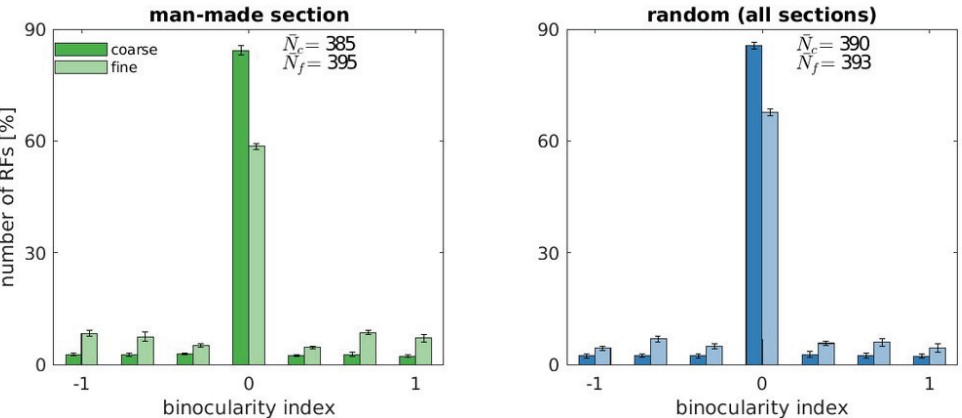

**Appendix 1—figure 2.** Binocularity values for five models trained with the man-made section or a random sample from the McGill database.

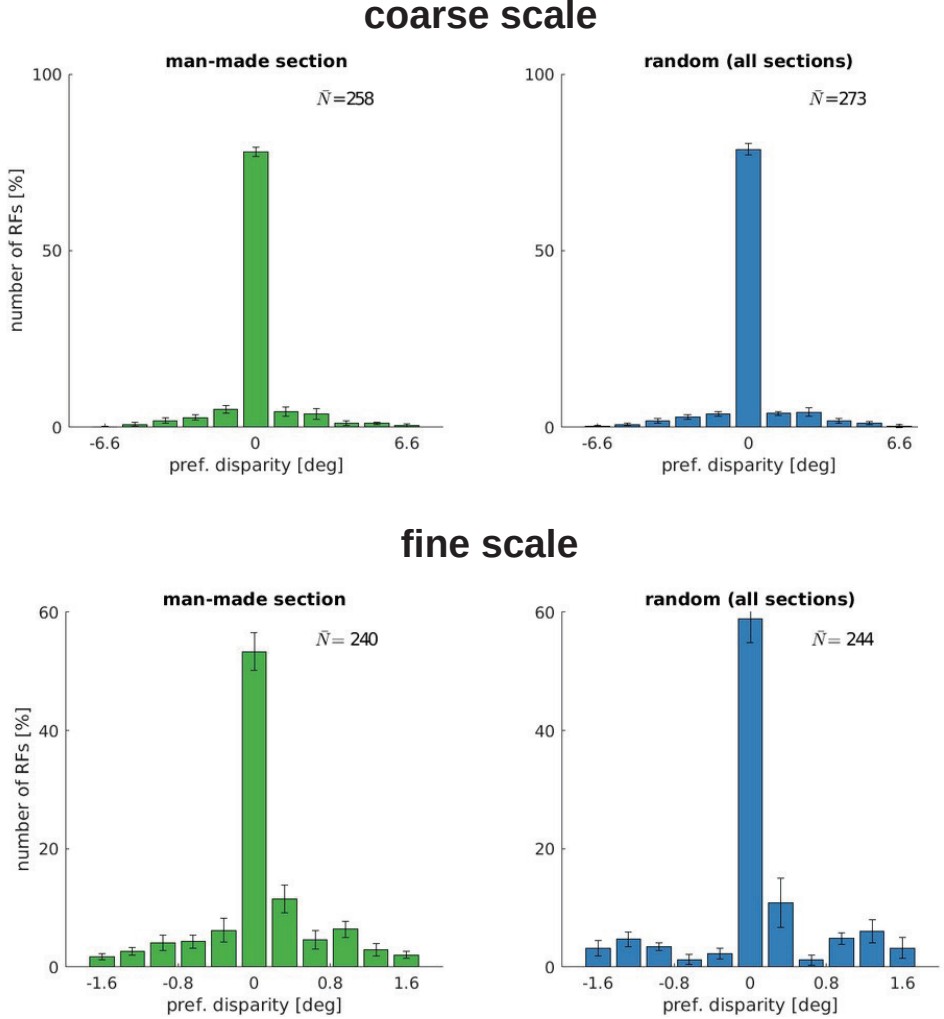

**Appendix 1—figure 3.** Disparity tuning for five models trained with the man-made section or a random sample from the McGill database.

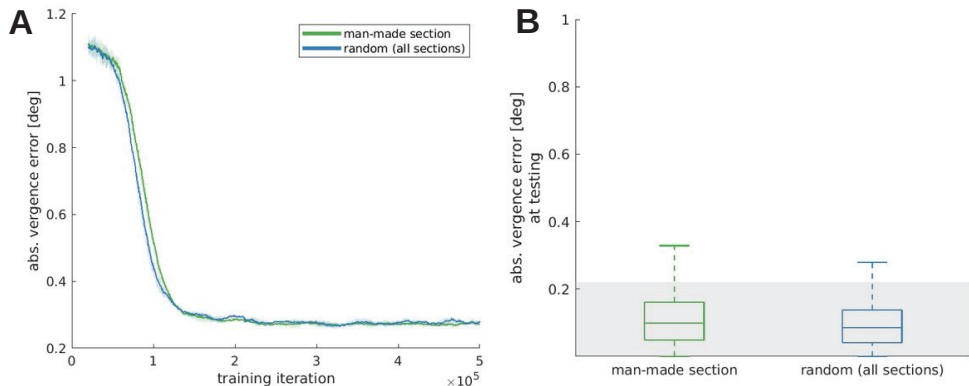

**Appendix 1—figure 4.** Vergence acuity over training time (A) and at testing (B) for five models trained with the man-made section or a random sample from the McGill database.

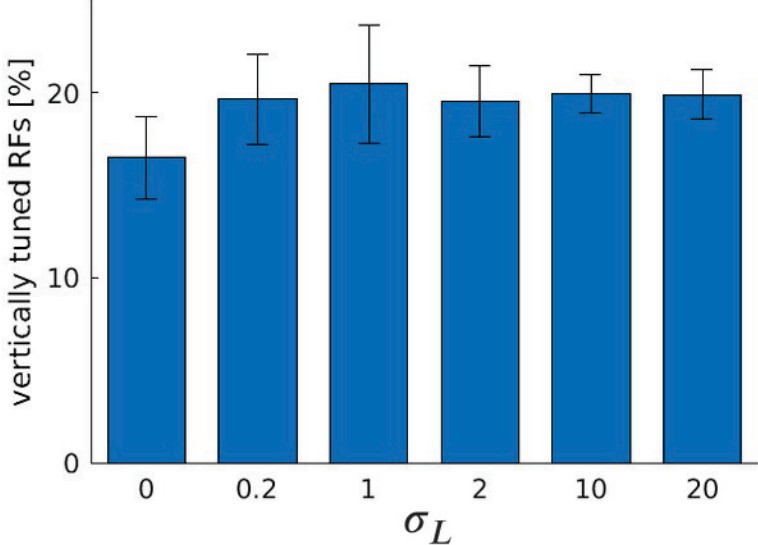

**Appendix 1—figure 5.** Number of RFs tuned to the vertical orientation for different values of $\sigma_L$, the standard deviation of the truncated Laplacian disparity input distribution, for five models trained with a random sample from the McGill database. Compared to the man-made section in *Figure 9B*, we observe an increase in the number of RFs for small but non-zero values of $\sigma_L$ and a decrease for bigger values of $\sigma_L$. However, the magnitude of the effect is reduced.

## Appendix 2

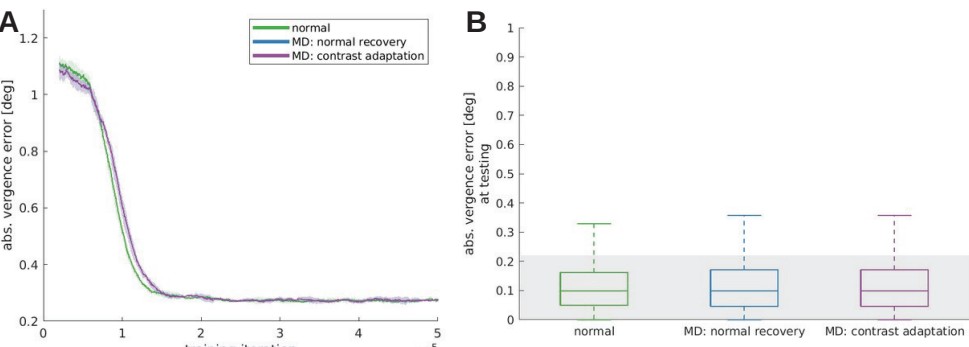

**Appendix 2—figure 1.** In response to the reviewers' comments, we tested the effect that patching the weak eye would have on the recovery from monocular deprivation (*Zhou et al., 2019*). To that end a model was trained under monocular deprivation, then normal visual input was reinstated and the weak eye received twice as much contrast as the other eye. This model, *MD constrast adaptation*, is compared to the *normal* and a reference model that did not receive an increased contrast (*MD normal recovery*), during training (**A**) and testing (**B**). All models trained under monocular deprivation can recover, when the RFs are still plastic. We do not observe a significant difference between the *normal recovery* and the *contrast adaptation*, probably because our model does not incorporate an interocular suppression mechanism that has been used to explain the effects of amblyopia on visual function (*Zhou et al., 2018*).

