## [Decision Letter]

**Acceptance summary:**

While traditional models of the development of the sensory cortices have relied on efficient coding principles and passive information processing, more recently active models have started to appear that minimize reconstruction error. In this paper Klimmasch et al., show that such model reproduce a number of experimental phenomena when applied to binocular vision.

**Decision letter after peer review:**

Thank you for submitting your article "The development of active binocular vision under normal and alternate rearing conditions" for consideration by *eLife*. Your article has been reviewed by 3 peer reviewers, one of whom is a member of our Board of Reviewing Editors, and the evaluation has been overseen by Joshua Gold as the Senior Editor. The following individual involved in review of your submission has agreed to reveal their identity: Laurent Perrinet (Reviewer #3).

The reviewers have discussed the reviews with one another and the Reviewing Editor has drafted this decision to help you prepare a revised submission.

Summary:

This paper demonstrates that the Active Efficient Coding (AEC) model that Shi and Triesch have presented in several previous papers is also able to reproduce standard results regarding the effects of abnormal visual experience.

Essential revisions:

1. There seems to be quite a bit of overlap in this study and your recent PNAS paper ("Active efficient coding explains the development of binocular vision and its failure in amblyopia"). It is critical for you to provide a strong, detailed explanation of the key differences between the two studies and how the present work represents substantially new insights.

2. As we understand, the action perception loop is "closed" by enacting the differential change in eye movements to the simulator. It is therefore different from other strategies like Active Inference for which a feedback prediction signal may be sent from motor to sensory systems. This difference should be highlighted in the text. Also, we are surprised that the study uses a very elaborate simulation system but that you simply show a *planar image of a natural image* and not a three-dimensional scene. Do you observe vergence movements if you scan the image? Finally, concerning the experimental procedure it is specified line 100 that "The plane is positioned in front the agent at variable distances" but no detail is given until Figure 8B (or we missed something).

3. A simple mathematical treatment could likely predict some results you have found, for instance the property that the network detects the right vergence for RDS. As a binary white noise, one can predict the distribution of activated coefficients for different disparities, especially given the kind of fits you perform on the basis functions. Also some elements on the bibliography of modeling emergence of binocular disparity (e.g., doi: 10.1523/JNEUROSCI.1259-18.2018) are lacking. These points should be clarified, including possibly de-emphasizing the RDS results if their novelty and importance to the overall message cannot be better established.

4. Some details lack in the presentation of the model, in particular concerning sparse coding (lines 474-488 page 15). Indeed, you explain the sparse coding scheme, but not how you implement sparse Hebbian learning. Either more details (e.g., equations) or references to existing work are needed. Also, how do you control the equalization of the coefficients? The pooling procedure generates some form of histogram (marginalization over all positions) and it is lacking in figure 1 and the generality of this procedure should be illustrated, for instance by comparing it to other schemes using histograms of features to classify images.

5. A nice extension of this model would be to predict how one could recover disparity acuity after abnormal rearing such as strabismus. For instance, it has been shown that patching the weak eye could be more effective that the usual strategy of patching the dominant eye (see Zhou, J, Reynaud, A., Yao, Z., Liu, R., Feng, L., Zhou, Y. and Hess, R. F. (2018) Amblyopic suppression: passive attenuation, enhanced dichoptic masking by the fellow eye or reduced dichoptic masking by the amblyopic eye?. Investigative Ophthalmology and Visual Science, 59, 4190-4197. for instance). Could you predict such a methodology? Predict another one?

---

## [Author Response]

Essential revisions1. There seems to be quite a bit of overlap in this study and your recent PNAS paper ("Active efficient coding explains the development of binocular vision and its failure in amblyopia"). It is critical for you to provide a strong, detailed explanation of the key differences between the two studies and how the present work represents substantially new insights.

The PNAS paper [Eckmann et al., (2019)] extends previous Active Efficient Coding (AEC) models that learn disparity tuning and vergence control to the learning of accommodation control. It also includes an interocular suppression mechanism to capture the role of interocular suppression for the development of amblyopia in children suffering from anisometropia. In the present paper, in contrast, we simulate a large range of alternate rearing conditions used in animal experiments. This allows us to test the AEC approach against a large body of neurophysiological work. For example, we compare neuronal tuning properties observed experimentally with those developing in the model. Specifically, we provide a detailed analysis of the role of behavior for the development of the statistics of orientation and disparity tuning as well as binocularity. In terms of the model itself, a major difference lies in the utilized reinforcement learning model. While Eckmann et al., (2019) used a very simplistic approach with small numbers of discrete actions for vergence and accommodation control, we here use a more sophisticated continuous action reinforcement learning model and a biomechanical model of the extra-ocular eye muscles, enhancing the realism of the model. Finally, in the current work we also test the model on random dot stereograms (RDSs), the most challenging kind of stereoscopic input and demonstrate that the model can “solve" RDSs, despite never being trained on them. Finally, we also study the phenomenon of aniseikonia and how it interferes with stereoscopic vision. We have brought another expert on board, Maria Fronius, to advise us on various aspects of the development of binocular vision from biological and clinical perspectives. We have clarified all these differences in the Introduction.

2. As we understand, the action perception loop is "closed" by enacting the differential change in eye movements to the simulator. It is therefore different from other strategies like Active Inference for which a feedback prediction signal may be sent from motor to sensory systems. This difference should be highlighted in the text.

Yes, it is correct that the action-perception loop is closed through motor actions (vergence eye movements) changing the next sensory input. The model does not use an explicit prediction of the next sensory input based on an efference copy of the motor command. It turns out that this is not necessary for the system to learn to perform at subpixel accuracy. It is an open question if such a mechanism might speed up learning. In active inference, an agents acts to make sensory inputs match prior expectations. But it is not always clear where these prior expectations come from. We have addressed these points in the Discussion.

Also, we are surprised that the study uses a very elaborate simulation system but that you simply show a planar image of a natural image and not a three-dimensional scene. Do you observe vergence movements if you scan the image? Finally, concerning the experimental procedure it is specified line 100 that "The plane is positioned in front the agent at variable distances" but no detail is given until Figure 8B (or we missed something).

The planar object surface allows us to precisely define the ground-truth for the vergence angle desired to fixate an object. If different points on a presented object have different distances from the observer, then the correct vergence angle in no longer uniquely defined and we wanted to avoid this. It is important to point out, however, that the model does not require planar objects and we have shown in previous work that AEC models can work well in more complex naturalistic 3D scences [Zhu et al., (2017), Lelais et al., (2019)].

More detail was added to the Results and Methods.

3. A simple mathematical treatment could likely predict some results you have found, for instance the property that the network detects the right vergence for RDS. As a binary white noise, one can predict the distribution of activated coefficients for different disparities, especially given the kind of fits you perform on the basis functions.

Indeed, given Gabor-fitted receptive fields and white noise inputs it should be possible to analytically calculate the average activation of our simulated simple cells. However, we do not think that this would provide any deeper insights into the learning processes, on which we are focusing here. Our primary concern is how the receptive fields develop their binocular tuning properties for natural input, how this input and the emerging vergence behavior determine the population statistics of receptive fields and how the neural activities get mapped to appropriate behavior (vergence commands), which in turn changes the input statistics and so forth. Therefore, we think it is better not to clutter the (already long) manuscript with an additional analysis like this, although it could certainly be done if required.

Also some elements on the bibliography of modeling emergence of binocular disparity (e.g., doi: 10.1523/JNEUROSCI.1259-18.2018) are lacking. These points should be clarified, including possibly de-emphasizing the RDS results if their novelty and importance to the overall message cannot be better established.

More references were added to the Introduction describing previous work on the development of binocular disparities in the context of different computational modeling frameworks. The RDS section was also adapted.

4. Some details lack in the presentation of the model, in particular concerning sparse coding (lines 474-488 page 15). Indeed, you explain the sparse coding scheme, but not how you implement sparse Hebbian learning. Either more details (e.g., equations) or references to existing work are needed. Also, how do you control the equalization of the coefficients? The pooling procedure generates some form of histogram (marginalization over all positions) and it is lacking in figure 1 and the generality of this procedure should be illustrated, for instance by comparing it to other schemes using histograms of features to classify images.

We added a new section to our Methods, containing formulae and additional references. Our pooling procedure is analogous to the operation of complex cells that pool responses of multiple simple cells across some region of visual space. This is also similar to approaches that utilize feature histograms to extract position-invariant features, for example to classify objects [Swain and Ballard (1991), Mel (1997)]. In these studies, it is common to normalize the coefficients/features in the histograms to make up for different sampling rates, different lighting conditions, etc. Here, we do not need to normalize the pooled values, because in our case, there is a fixed number of active features (10) per image patch. To clarify all this, we added an explanatory section to the Methods.

To keep the simplicity of Figure 1, we did not include the pooling procedure here.

5. A nice extension of this model would be to predict how one could recover disparity acuity after abnormal rearing such as strabismus. For instance, it has been shown that patching the weak eye could be more effective that the usual strategy of patching the dominant eye (see Zhou, J, Reynaud, A., Yao, Z., Liu, R., Feng, L., Zhou, Y. and Hess, R. F. (2018) Amblyopic suppression: passive attenuation, enhanced dichoptic masking by the fellow eye or reduced dichoptic masking by the amblyopic eye?. Investigative Ophthalmology and Visual Science, 59, 4190-4197. for instance). Could you predict such a methodology? Predict another one?

Thank you for this suggestion, which motivated us to do an additional analysis. We have assumed that patching the weak eye results in a boost of the contrast sensitivity after removing the patch. To capture this in our model, we increased the contrast in the deprived eye of a monocularly reared model after normal visual input was reinstated. However, we saw no difference to a model that learned without this contrast adaptation. We can not rule out, however, that this could change if we included an interocular suppression mechanism to the model. We added these new results to the manuscript (see Appendix 2).

As another clinically relevant aspect, we also added experiments on a common side effect of the treatment of anisometropia: Correction of the refraction differences between the eyes can lead to different magnifications of the input images on the patients' retinas, a condition called aniseikonia. We show that aniseikonia impedes visual development and can lead to a complete loss of binocular function. Interestingly, we also find that the number of cells tuned to vertical disparities increases with increasing aniseikonia (until some degree), which represents another testable prediction of our model.